# Calibrated Knowledge Aggregation in Bayesian Mixture-of-Experts for Continual VQA

**Mahsa Mozaffari**[1]  **Hitesh Sapkota**[2]  **Yu Kong**[3]  **Xumin Liu**[1]  **Qi Yu**[1 2]

## Abstract

Continual learning for visual question answering (VQA) is typically implemented by training one expert per task and routing each query using task-ID supervision. Yet continual VQA tasks overlap substantially: on the VQA-v2 task stream, a non-native expert outperforms the task's own expert on $49.9\%$ of queries, so hard routing both wastes transferable knowledge and can be confidently wrong when mismatched. We propose a calibrated Bayesian mixture-of-experts that trains parameter-efficient per-task adapters, learns routing by directly maximizing expected VQA utility, and marginalizes expert identity at inference via Bayesian aggregation in a unified answer space; an entropy penalty prevents the utility objective from collapsing to one-hot routing, enabling evidence pooling across plausible experts. We reach 64.16 accuracy with 0.63 forgetting on VQA-v2 CL-LS (+5.74% accuracy, -2.99 forgetting vs. the strongest prior method), 78.81 with 0.40 forgetting on TDIUC CL-LS (+3.10, -1.74), and 83.41 with 3.21 forgetting on TDIUC CL-VS (+1.58, -0.82). Calibration also improves on VQA-v2, reducing ECE from 0.15 to 0.07.

## 1. Introduction

Continual learning (CL) seeks to expand a model's capabilities by learning tasks sequentially without retraining from scratch or catastrophically forgetting earlier knowledge (Kirkpatrick et al., 2017; Lopez-Paz & Ranzato, 2017; Chaudhry et al., 2018). This setting is increasingly relevant for visual question answering (VQA), where systems must adapt as new question types, visual domains, annotation

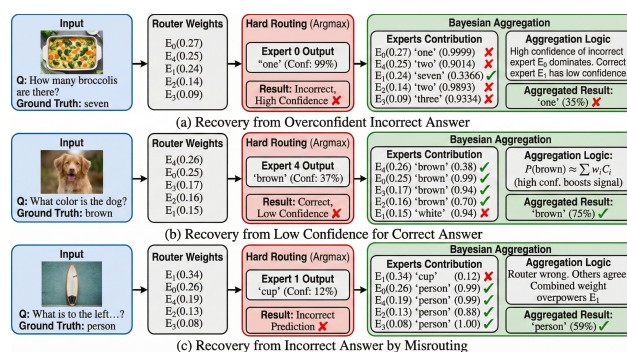

*Figure 1.* Our calibrated Bayesian aggregation in a mixture-of-experts model improves reliability over hard (top-1 expert) routing by combining evidences of expert predictive distributions to recover from (a) overconfident wrong predictions, (b) underconfident correct predictions, (c) incorrect predictions due to misrouting.

conventions, and datasets appear. In practical deployments, including interactive assistants (Gordon et al., 2018; Gurari et al., 2018), embodied agents (Das et al., 2018), and decision-support pipelines (Lau et al., 2018), knowing *when to trust* a prediction can be as important as the prediction itself. Yet modern VQA models are frequently miscalibrated, especially under distribution shift (Guo et al., 2017; Naeini et al., 2015; Ovadia et al., 2019); continual VQA exacerbates this issue because each new task introduces additional sources of shift and out-of-distribution (OOD) queries.

A common continual-VQA recipe is modularity: allocate task-specific and parameter-efficient adapters while keeping most shared parameters fixed (Hu et al., 2021; Wang et al., 2022b; Qian et al., 2023). Mixture-of-experts (MoE) architectures provide a natural modular instantiation: one expert per task, and a gating network that selects which expert(s) to use at inference (Jacobs et al., 1991; Shazeer et al., 2017). In many CL benchmarks, routing is supervised by task identity and inference is *hard* (route to a single expert). This implicitly assumes tasks are well-separated and that the "correct task expert" is the uniquely appropriate solver.

Our first key observation is that **continual VQA violates this separability assumption**. Tasks overlap substantially in *input space* (shared images and visual concepts) and in *output space* (overlapping answer sets). As a result, non-native experts often retain non-trivial competence on other tasks. Figure 2a visualizes this effect as a cross-task transfer

---

[†]Work not related to Amazon.

[1]Rochester Institute of Technology, Rochester, NY, USA [2]Amazon, USA[†] [3]Michigan State University, East Lansing, MI, USA. Correspondence to: Qi Yu <qi.yu@rit.edu>.

*Proceedings of the 43rd International Conference on Machine Learning*, Seoul, South Korea. PMLR 306, 2026. Copyright 2026 by the author(s).

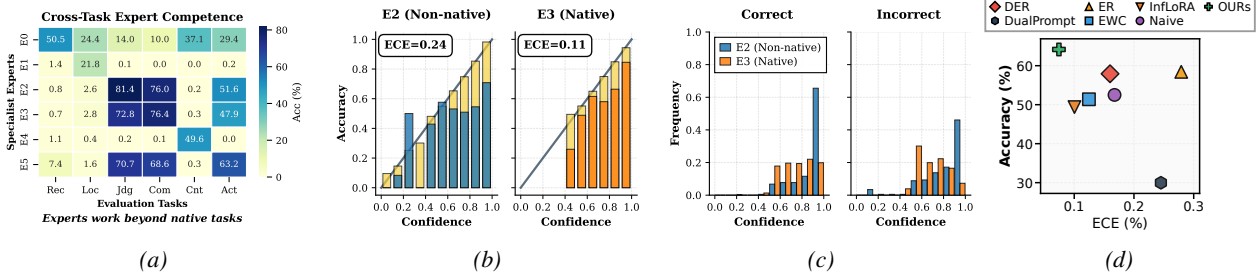

*Figure 2.* (a) Cross-task competence of experts on VQA-v2. Rows correspond to experts `E0-E5` (each trained on its native question-type task); columns correspond to tasks at test time. Diagonal blocks reflect in-task accuracies, many off-diagonal values are non-trivial. (b-c) reliability diagrams and correct/incorrect confidence distribution of native& non-native experts on commonsense task. (d) Accuracy vs. calibration comparison of our method with baselines.

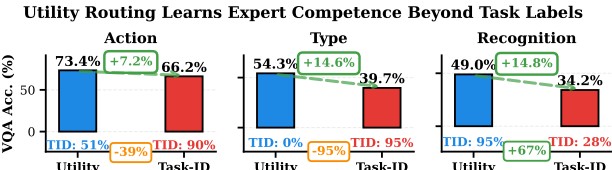

*Figure 3.* Per-task comparison of utility-trained vs. task-ID supervised routers. **Action & Type:** Utility routing achieves higher VQA accuracy (+7.2%, +14.6%) by routing to *non-native* experts, (lower task-ID accuracy) but *better answers*. **Recognition:** Utility routing achieves higher VQA accuracy (+14.8%) by more consistently selecting the *native* expert (+67% task-ID accuracy). Utility routing learns which experts are actually competent for each query, not just which match nominal task labels.

matrix with many strong off-diagonal entries. Moreover, Table 1 shows that for a non-negligible fraction of queries, a non-native expert outperforms the oracle "correct-task" expert (the one indexed by ground-truth task ID). These results imply that even *perfect* task-ID routing can be suboptimal. In continual VQA, routing should be treated as capability selection (which expert is useful for this input), not merely task identification (which dataset generated this input).

This motivates routing objectives that directly target downstream VQA performance. Instead of training a router to predict task ID, or using proxies such as autoencoder reconstruction error (Aljundi et al., 2017), we train the router to place higher weight on experts that are likely to answer correctly for the current input. Concretely, we learn soft routing weights by maximizing expected VQA utility under each expert's predictive distribution, so the router is rewarded for selecting experts that produce high expected score on the query. This reframes routing as the mechanism that unlocks cross-task reuse: when tasks overlap, the router can exploit transferable knowledge by assigning probability mass to non-native but competent experts (see Figure 3).

A second, equally important challenge is that *expert competence does not imply expert reliability*. Neural networks are often overconfident on out-of-distribution inputs (Guo et al., 2017; Ovadia et al., 2019), and in a modular continual setting, an expert is frequently evaluated on queries that differ from its training distribution. In continual VQA this matters

*Table 1.* Non-native expert outperformance (%) on VQA-v2 tasks.

| Action | Causal | Color | C.sense | Count | Judge | Loc. | Recog. | Subcat. | Type | **Avg** |
|--------|--------|-------|---------|-------|-------|------|--------|---------|------|---------|
| 57.3 | 45.2 | 93.5 | 27.5 | 19.7 | 30.4 | 62.5 | 58.6 | 29.3 | 75.5 | **49.9** |

precisely because cross-task reuse is common: the router may correctly discover that a non-native expert can answer a query well, yet that expert can still be poorly calibrated on that query distribution. This failure mode is especially damaging when inference commits to a single expert (hard routing), where a mismatch can produce confidently wrong predictions. We empirically illustrate this failure mode in Figure 2b, which demonstrate the reliability diagrams of native (right) vs. non-native (left) experts. Despite their similar accuracy of 76% on the 'commonsense' task, the non-native expert (E2 trained on Judge), is overconfident, and produces poorly calibrated confidences, compared to the native expert (E3). This is further evidenced by Fig. 2c, which compares the confidence distributions of correct and incorrect answers for both experts. As the figure demonstrates, the non-native expert shows a large concentration of incorrect answers with confidences close to 1.

To address both shareable expertise and miscalibrated confidence, we propose a framework for calibrated knowledge aggregation in continual VQA. Our approach couples (i) an MoE CL backbone with task-specific experts for mitigating forgetting, with (ii) a utility-trained router that learns meaningful expert weights, and (iii) a calibrated aggregation mechanism that combines expert predictions to reduce overconfidence. Importantly, naively training a utility-based router produces overly sharp expert posteriors, effectively collapsing into hard routing and limiting the benefits of multi-expert sharing. This motivates explicitly calibrating the router's uncertainty (e.g., via entropy-promoting regularization), enabling aggregation between top experts to meaningfully incorporate multiple experts and yielding improved end-to-end calibration (lower Expected Calibration Error (ECE)) without degrading accuracy (Figure 2d).

Our empirical study shows that calibrated knowledge sharing improves continual VQA along two axes: (1) robustness to forgetting / improved accuracy via utility-aware routing,

and (2) improved probabilistic reliability via calibrated expert aggregation. Across continual VQA benchmarks, we observe that our method maintains or slightly improves final accuracy while substantially reducing miscalibration, primarily by lowering confidence on incorrect predictions.

Concretely, this paper makes the following contributions:

- **Problem framing:** We identify and quantify a distinctive property of continual VQA: experts generalize non-trivially beyond their native tasks, and non-native experts can outperform the oracle task expert on a meaningful subset of queries (Figure 2a, Table 1).
- **Utility-driven routing:** We introduce utility-based router training for continual VQA that directly optimizes expected downstream performance, improving over task-ID-supervised routing in both average accuracy and forgetting-related metrics.
- **Calibrated knowledge sharing:** We show that naïve routing objectives can induce sharp, near one-hot expert weights, limiting knowledge sharing, and we propose router calibration to maintain informative uncertainty over experts (Figure 5a).
- **Comprehensive evaluation:** We demonstrate that calibrated expert aggregation improves end-to-end VQA calibration (ECE) by reducing overconfident wrong answers while preserving accuracy (Figure 2d).

Overall, our results suggest that the central opportunity in continual VQA is not merely selecting the "right" expert by task identity, but *sharing the right knowledge in a calibrated way* across experts when task boundaries are inherently ambiguous.

## 2. Related Work

**VQA and vision–language models.** Modern VQA is largely built on large-scale vision–language pretraining with Transformer encoders, including two-stream and single-stream designs (e.g., ViLBERT/LXMERT, UNITER) (Lu et al., 2019; Tan & Bansal, 2019; Chen et al., 2020), as well as more efficient end-to-end variants (e.g., ViLT) and improved alignment objectives (e.g., ALBEF, FLAVA) (Kim et al., 2021; Li et al., 2021a; Singh et al., 2022). Prompted and instruction-tuned VLMs further strengthen VQA performance but introduce additional reliability considerations tied to generative behavior (Li et al., 2023; Liu et al., 2023). Beyond accuracy, VQA is known to exhibit shortcut learning and sensitivity to dataset biases (Agrawal et al., 2018; Zhang et al., 2023a), and calibration under distribution shift has become a central evaluation axis (Guo et al., 2017; Ovadia et al., 2019; Whitehead et al., 2022).

**Continual learning.** Continual learning (CL) mitigates catastrophic forgetting via replay, regularization, or architectural isolation/expansion (Wang et al., 2024; Kirkpatrick et al., 2017; Buzzega et al., 2020). For large pretrained back-bones, parameter-efficient CL (adapters/prompts/LoRA) has become common because it limits interference while preserving foundation-model priors. In this context, modular and conditional-computation approaches—especially mixture-of-experts (MoE)—are a natural fit, using routing/gating to control specialization and reuse (Shazeer et al., 2017; Aljundi et al., 2017; Fernando et al., 2017). Recent MoE-style CL for (vision-)language models explicitly studies routing under shift and task-free expert assignment (Yu et al., 2024; 2025; Julian et al., 2025; Thérien et al., 2025).

**Continual VQA.** Continual VQA has recently gained attention through benchmarks (e.g., VQACL (Zhang et al., 2023b), CL-CrossVQA (Zhang et al., 2025)) and methods spanning prompt learning, pseudo-rehearsal, and distillation/replay tailored to multimodal drift (Zhang et al., 2023b; 2025; Qian et al., 2023; Das et al., 2024; Nikandrou et al., 2024). Several works also explore modularization for continual VQA (including MoE-style designs) (Huai et al., 2025; Li & Lyu, 2025). However, many modular approaches still effectively treat routing as task identification and/or rely on hard expert choice, which can underutilize cross-task overlap and amplify overconfident errors when routing is imperfect. Our work targets this gap by training routing for *expected downstream VQA utility* (rather than task-ID), and by performing *Bayesian aggregation* in a shared answer space with entropy-regularized routing to preserve meaningful uncertainty and improve probabilistic reliability.

## 3. Preliminaries

**Formulation of continual VQA.** A VQA task involves predicting an answer $a$ to a natural language question $q$ given an input image $v$. We formulate VQA as a classification task, where the answers are selected from a predefined answer set. Given a dataset $\mathcal{D} = \{(v_n, q_n, a_n)\}_{n=1}^{N}$, with images $v_n$, questions $q_n$, and corresponding answers $a_n$, the model learns a function $f_\Theta : \mathcal{V} \times \mathcal{Q} \to \mathcal{Y}$ parameterized by $\Theta$, where $\mathcal{V}$, $\mathcal{Q}$, and $\mathcal{Y}$ represent the space of images, questions, and possible answers, respectively. In VQA, answer space refers to the set of all possible answers, in dataset.

In the CL setting for VQA, the model learns from tasks $\mathcal{T}_1, \mathcal{T}_2, \ldots, \mathcal{T}_T$, in sequence. At each time step $t$, the model trains on task $\mathcal{T}_t$ without access to past datasets (except for a limited memory buffer $\mathcal{B}$ of size $M$ storing past examples). Each task has dataset $\mathcal{D}_t$ and answer space $\mathcal{Y}_t$, where $\bigcup_{t=1}^{T} \mathcal{Y}_t = \mathcal{Y}$. As shown in Figure 8, VQA's complexity results in varying overlap between $\mathcal{Y}_t$, making continual VQA challenging. We study task-agnostic continual VQA, where task boundaries are available during training, but the task identity is *unknown* at test time. Additionally, answer spaces across tasks can be shared, or partially overlapping.

**Mixture-of-Expert for continual learning.** Mixture-of-Experts (MoE) (Jacobs et al., 1991) for CL employs multiple specialized subnetworks (experts) each trained to handle dif-

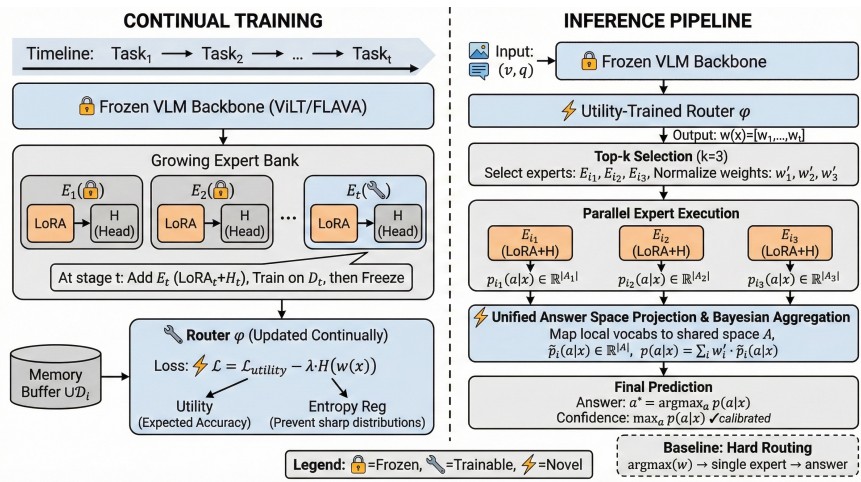

*Figure 4.* The overall workflow of our Continual VQA framework

ferent tasks or data distributions, with a gating mechanism that dynamically routes the inputs to the experts. MoE help mitigate catastrophic forgetting by allocating new knowledge to distinct experts while preserving previously acquired knowledge through other experts. The core concept of MoE is formalized as:

$$f_{\text{MoE}}(\mathbf{x}; \{\Theta^{(t)}\}_{t=1}^T, \varphi) = \sum_{t=1}^T w_\varphi(t \mid \mathbf{x}) \cdot f_t(\mathbf{x}; \Theta^{(t)}) \quad (1)$$

where $f_t$ is the $t$-th expert, $\Theta^{(t)}$ are expert-specific parameters, $w_\varphi(t \mid \mathbf{x})$ is the gating weight for $t$-th expert with input $\mathbf{x}$ and parameter $\varphi$, and $T$ is the number of experts.

LoRA (Hu et al., 2022) enables a parameter-efficient MoE by injecting a tiny set of trainable low-rank matrices into a frozen, pretrained backbone. For a frozen pre-trained weight matrix $W \in \mathbb{R}^{d \times k}$, LoRA parameterizes the update as: $W' = W + \Delta W = W + BA$, where $B \in \mathbb{R}^{d \times r}$ and $A \in \mathbb{R}^{r \times k}$ are low-rank matrices with $r \ll \min(d, k)$, significantly reducing trainable parameters compared with full model training. LoRA-based MoE approaches (Yu et al., 2024) combine parameter-efficient low-rank adaptation with MoE, updating only lightweight LoRA parameters instead of full weights. Freezing pre-trained weights preserves pre-trained knowledge crucial for foundation models. Modular expert design enables task-specific adaptations, preventing forgetting. Gating strategies range from hard routing based on task ID (Yu et al., 2024) to learned soft mechanisms that adaptively combine expert outputs (Han et al., 2024).

**VQA Calibration.** For an input $\mathbf{x}$, the model outputs a distribution over answers $p_\Theta(a|\mathbf{x})$. We denote the predicted answer $\hat{a}(\mathbf{x}) = \arg\max_a p_\Theta(a|\mathbf{x})$ and its confidence $c(\mathbf{x}) = p_\Theta(\hat{a}(\mathbf{x})|\mathbf{x})$. Let $s(\mathbf{x}) \in \{0, 1\}$ denote the per-example VQA score under the standard VQA evaluation protocol. A predictor is *calibrated* if $\mathbb{E}[s(\mathbf{x}) \mid c(\mathbf{x}) = q] = q$ for all $q \in [0, 1]$. We summarize deviations from this ideal using Expected Calibration Error (ECE).

## 4. Methodology

The overall framework of our method is shown in Figure 4. We formulate continual VQA as a mixture of frozen, task-specialized experts combined by a learned router. At each stage $t \in [T]$, the method proceeds in three steps: *(1) Per-task expert construction (training time).* Starting from a pretrained vision–language backbone $F_{\theta_0}$ that remains frozen throughout, we instantiate a new lightweight expert by training only a task-specific LoRA adapter $\Delta\theta_t$ and a task-specific answer head $H_t$ on $\mathcal{D}_t$. After training, $(\Delta\theta_t, H_t)$ are frozen and added to the expert pool $\mathcal{E}_{1:t}$, preventing forgetting while enabling modular growth. *(2) Utility-driven router training (training time).* Given the frozen experts $\mathcal{E}_{1:t}$, we train a lightweight router $w_\varphi(\cdot \mid \mathbf{x})$ on a mixture of current-task data and replayed examples $(\mathcal{D}_t \cup \mathcal{B})$ to predict *which expert(s) are useful for the input*, rather than the generating task. Concretely, the router is optimized to maximize the expected downstream VQA utility of the induced mixture, and we regularize its entropy to avoid degenerate near one-hot solutions. *(3) Bayesian inference by top-k aggregation (test time).* For a test input $x$, the router produces mixture weights. Because tasks may have different answer vocabularies, each expert distribution is first mapped into a shared answer space (the union of all observed answers), and we then form the final predictive distribution by Bayesian aggregation, $p(a \mid \mathbf{x}) = \sum_t w_\varphi(t \mid \mathbf{x}) p_t^{\text{unified}}(a \mid \mathbf{x})$, returning $\arg\max_a p(a \mid \mathbf{x})$ and its confidence.

### 4.1. Parameter-Efficient Experts

We build on a pretrained vision-language backbone $F_{\theta_0}$ that remains *frozen* throughout continual learning. For each incoming task $\mathcal{T}_t$, we instantiate a task expert consisting of LoRA adapters $\Delta\theta_t = \{B_i^{(t)} A_i^{(t)}\}_{i=1}^L$ injected into $L$ backbone layers and a task-specific linear head $H_t$. Given an input $\mathbf{x} = (v, q)$, the task-adapted backbone produces a representation $h_t(\mathbf{x}) = F_{\theta_0, \Delta\theta_t}(\mathbf{x})$, and the task head

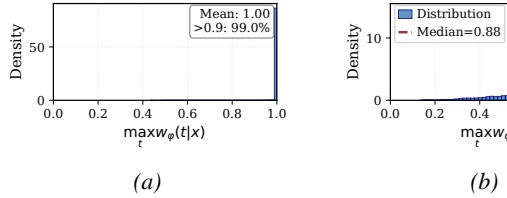

*Figure 5.* Maximum router weight distributions in (a) without entropy regularization, (b) entropy-regularized utility-trained router.

outputs task-local logits $\ell_t(\mathbf{x}) = H_t\big(h_t(\mathbf{x})\big) \in \mathbb{R}^{|\mathcal{Y}_t|}$. We convert logits to a task-local answer distribution via

$$p_t^{\mathrm{loc}}(a \mid \mathbf{x}) = \mathrm{softmax}(\ell_t(\mathbf{x}))_a, \qquad a \in \mathcal{Y}_t. \quad (2)$$

For cross-task aggregation, we map $p_t^{\mathrm{loc}}(\cdot \mid \mathbf{x})$ into a shared answer space $\mathcal{Y}$ (Sec. 4.4) to obtain $p_t(\cdot \mid \mathbf{x})$. After training on $\mathcal{D}_t$, the expert parameters $(\Delta\theta_t, H_t)$ are frozen and added to the expert set $\mathcal{E}_{1:t}$.

## 4.2. Utility-driven Router Training

At continual stage $t$, for an input $\mathbf{x} = (v, q)$, each expert in $\mathcal{E}_{1:t} = \{E_1, \ldots, E_t\}$, produces a predictive distribution $p_t(\cdot \mid \mathbf{x})$ over the *shared* answer space $\mathcal{Y}$ (Sec. 4.4). The router is a lightweight network $r_\varphi(\mathbf{x}) \in \mathbb{R}^t$ that outputs mixture weights

$$w_\varphi(t \mid \mathbf{x}) = \mathrm{Softmax}_t\big(r_\varphi(\mathbf{x})\big), \qquad w_\varphi(\cdot \mid \mathbf{x}) \in \Delta^{t-1}, \quad (3)$$

where $\Delta^{t-1}$ is the probability simplex.

Rather than supervising the router with task-ID labels, we train it to predict *which expert(s) are most useful for the current input*. Let $\mathcal{A}(\mathbf{x})$ denote the VQA annotations for $\mathbf{x}$, and define the standard VQA soft score for an answer $a \in \mathcal{Y}$ as

$$s_\mathbf{x}(a) \triangleq \mathrm{score}(a; \mathcal{A}(\mathbf{x})) \in [0, 1]. \quad (4)$$

We define the *utility* of expert $t$ on $\mathbf{x}$ as its expected VQA score under its predictive distribution:

$$U_t(\mathbf{x}) \triangleq \mathbb{E}_{a \sim p_t(\cdot \mid \mathbf{x})}\big[s_\mathbf{x}(a)\big] \quad (5)$$
$$= \sum_{a \in \mathcal{Y}} p_t(a \mid \mathbf{x}) \, s_\mathbf{x}(a) = \langle p_t(\cdot \mid \mathbf{x}), s_\mathbf{x}(\cdot) \rangle.$$

Given frozen experts, at stage $s$, the router is trained to maximize the expected utility of the induced mixture:

$$\max_\varphi \; \mathbb{E}_{\mathbf{x} \sim \mathcal{D}_s \cup \mathcal{B}}\left[ \sum_{t=1}^s w_\varphi(t \mid \mathbf{x}) \, U_t(\mathbf{x}) \right], \quad (6)$$

where $\mathcal{B}$ is a replay buffer of past-task examples, ensuring the router remains sensitive to prior experts as tasks arrive.

**Utility routing tends to become sharp.** For a fixed input $\mathbf{x}$, the inner objective in eq. (6) is *linear* in $w$ and constrained to the simplex: $\max_{w \in \Delta^{s-1}} \sum_t w_t \, U_t(\mathbf{x})$. A linear objective over a simplex achieves its maximum at an *extreme*

*point*, i.e., a one-hot vector that selects $\arg\max_t U_t(\mathbf{x})$. Thus, absent additional regularization, the utility objective *promotes near-deterministic routing*. Figure 5a provides empirical confirmation that the learned $w_\varphi(\cdot \mid \mathbf{x})$ indeed becomes highly concentrated.

## 4.3. Calibrating the Router via Entropy Regularization

Utility routing in eq. (6) is performance-aligned, but the simplex/linearity argument above implies it naturally drives $w_\varphi(\cdot \mid \mathbf{x})$ toward extreme (one-hot) solutions, which limits its multi-expert evidence pooling in Bayesian aggregation (Sec. 4.4) and can amplify overconfident errors. We therefore use an entropy-regularized objective that favors *high-utility yet non-degenerate* mixtures:

$$\max_\varphi \; \mathbb{E}_{\mathbf{x} \sim \mathcal{D}_s \cup \mathcal{B}}\left[ \sum_{t=1}^s w_\varphi(t \mid \mathbf{x}) \, U_t(\mathbf{x}) + \lambda \, H\big(w_\varphi(\cdot \mid \mathbf{x})\big) \right],$$
$$\text{where,} \; H\big(w_\varphi(\cdot \mid \mathbf{x})\big) \triangleq - \sum_{t=1}^s w_\varphi(t \mid \mathbf{x}) \log w_\varphi(t \mid \mathbf{x}). \quad (7)$$

$\lambda$ controls sharpness (recovering hard selection as $\lambda \to 0$). Empirically, this regularization reduces routing concentration (Fig. 5b), enabling Bayesian aggregation to combine multiple competent experts and improving end-to-end confidence reliability.

## 4.4. Inference via Bayesian Aggregation of Experts

After learning stage $t$, we have a set of frozen experts $\mathcal{E}_{1:s} = \{E_1, \ldots, E_s\}$ and a router $w_\varphi(\cdot \mid x) \in \Delta^{s-1}$. At test time, we treat the expert index $z \in \{1, \ldots, s\}$ as a latent variable and perform Bayesian model averaging over experts. Given an input $\mathbf{x}$, the router provides mixture weights $w_\varphi(t \mid \mathbf{x}) \approx p(z = t \mid \mathbf{x})$, and each expert produces a predictive distribution $p_t(a \mid \mathbf{x})$ over a shared answer space $\mathcal{Y}$ (defined below). The aggregated predictive distribution is

$$p(a \mid x) = \sum_{t=1}^s w_\varphi(t \mid x) \, p_t(a \mid x), \qquad a \in \mathcal{Y}. \quad (8)$$

This reduces to hard routing when $w_\varphi(\cdot \mid x)$ is (near) one-hot. Crucially, eq. (8) preserves uncertainty over $z$: when experts disagree, the convex combination spreads mass across competing answers, typically reducing over-confident errors; when multiple experts agree, their evidence is pooled by summing probability mass on the same answer.

**Normalization.** Since $w_\varphi(\cdot \mid x)$ is a probability distribution and each $p_t(\cdot \mid x)$ is normalized, the mixture is also a valid probability distribution:

$$\sum_{a \in \mathcal{Y}} p(a \mid \mathbf{x}) = \sum_t w_\varphi(t \mid \mathbf{x}) \sum_{a \in \mathcal{Y}} p_t(a \mid \mathbf{x}) \quad (9)$$
$$= \sum_t w_\varphi(t \mid \mathbf{x}) = 1.$$

**Top-$k$ mixture for efficiency.** Evaluating all experts scales linearly with $s$. We therefore restrict aggregation to the top-$k$ experts under the router. Let $\mathcal{S}_k(\mathbf{x})$ be the indices of the top-$k$ weights in $w_\varphi(\cdot \mid \mathbf{x})$ and define the renormalized weights

$$\tilde{w}_\varphi(t \mid \mathbf{x}) = \frac{w_\varphi(t \mid \mathbf{x})}{\sum_{j \in \mathcal{S}_k(\mathbf{x})} w_\varphi(j \mid \mathbf{x})} \, \mathbb{1}\{t \in \mathcal{S}_k(\mathbf{x})\}. \quad (10)$$

We then compute $p_k(a \mid \mathbf{x}) = \sum_{t \in \mathcal{S}_k(\mathbf{x})} \tilde{w}_\varphi(t \mid \mathbf{x}) \, p_t(a \mid \mathbf{x})$. In practice, small $k$ captures most of the mixture mass when the router is moderately sparse.

**Answer-space alignment for aggregation.** Each task $\mathcal{T}_t$ has its own answer set $\mathcal{Y}_t$, so expert outputs are not directly comparable. To enable aggregation, we maintain a unified answer vocabulary over observed tasks: $\mathcal{Y} = \bigcup_t \mathcal{Y}_t$. Let $p_t^{\text{loc}}(\cdot \mid \mathbf{x})$ denote expert $t$'s distribution in its local space $\mathcal{Y}_t$. We precompute a sparse alignment map $M_t \in \{0, 1\}^{|\mathcal{Y}| \times |\mathcal{Y}_t|}$ that maps each local answer index to its unified index, and define the unified distribution by

$$p_t(\cdot \mid \mathbf{x}) = M_t \, p_t^{\text{loc}}(\cdot \mid \mathbf{x}). \tag{11}$$

Entries corresponding to answers not in $\mathcal{Y}_t$ are filled with 0, meaning expert $t$ contributes no probability mass to answers outside its vocabulary. Because each local answer maps to exactly one unified answer, eq. (11) preserves normalization: $\sum_{a \in \mathcal{Y}} p_t(a \mid \mathbf{x}) = \sum_{a \in \mathcal{Y}_t} p_t^{\text{loc}}(a \mid \mathbf{x}) = 1$. This alignment also ensures that overlapping answers across tasks, receive pooled evidence under Bayesian aggregation.

**Remark.** Our approach can be viewed as a Bayesian MoE for continual VQA: instead of committing to a single expert, we marginalize over the latent expert index and aggregate expert posteriors in a shared answer space. This is particularly important in continual VQA, where (i) tasks overlap in inputs and outputs and (ii) non-native experts can still be competent, so hard task-ID routing can produce overconfident errors. Aggregation pools evidence when experts agree and preserves uncertainty when they disagree, yielding a more reliable predictive distribution (Fig. 1). Crucially, effective aggregation requires a *calibrated* router: since the routing weights live on the probability simplex, optimizing utility alone tends to push solutions toward simplex vertices (i.e., near one-hot routing), collapsing back to single-expert behavior; entropy-based calibration counteracts this collapse by maintaining non-trivial posterior mass on plausible experts, enabling multi-expert sharing and improving confidence reliability (Fig. 5).

# 5. Experiments

We evaluate in a task-incremental setting: tasks arrive sequentially, and the model is updated after observing each task once. After learning task $t$, we evaluate on tasks seen so far. Task identities are only available during training, but *not at test time*.

## 5.1. Experimental Setup

**Backbones and parameter-efficient experts.** We instantiate a frozen vision–language backbone (ViLT and FLAVA) and allocate one expert per task. Each expert consists of task-specific LoRA adapters (rank $r=8$, injected into the FFN modules of all Transformer blocks) and a task-local answer head over $\mathcal{Y}_t$. At stage $t$, we train only the new expert on $\mathcal{D}_t$ (with standard VQA BCE training) and then freeze it, yielding a growing lightweight expert pool without modifying past experts.

**Router, replay, and inference.** The router is a lightweight two-layer MLP that takes a frozen question embedding (text-only) and outputs a distribution over experts. At stage $t$, we train the router on a mixture of current-task samples and replay from a memory buffer $\mathcal{B}$ (size $|\mathcal{B}|=5000$) to maintain sensitivity to older experts. At inference, we select the top-$k$ experts under the router ($k=3$ unless otherwise stated), renormalize their weights, and perform Bayesian aggregation in the unified answer space.

**Benchmarks.** We evaluate on (i) VQA-v2 (Antol et al., 2015) under the 10 question-type task stream of VQACL (Zhang et al., 2023b), and (ii) TDIUC (Qian et al., 2023) under the two continual protocols from TRIPLET (Qian et al., 2023): CL-LS (language/question-type shift) and CL-VS (visual-domain shift),

each with 5 tasks. Detailed task definitions and task overlap analysis are reported in Appendix F.

**Baselines.** We compare against: (i) generic continual-learning baselines: Naive fine-tuning, EWC (Kirkpatrick et al., 2017), ER (Chaudhry et al., 2018), DER (Buzzega et al., 2020), GEM (Lopez-Paz & Ranzato, 2017), (ii) parameter-efficient CL methods for pretrained VLMs: DualPrompt (Wang et al., 2022a), InfLoRA (Liang & Li, 2024), (iii) continual-VQA method: VQACL (Zhang et al., 2023b), adapted to our encoder-only backbones.

**Metrics.** We report Micro and Macro accuracy (sample and task level), Average Forgetting (AF), and calibration via Expected Calibration Error (ECE). Let $S_{i,j}$ denote task $i$ accuracy at stage $j$. Accuracies are:

$$\texttt{Macro} = \frac{1}{T} \sum_{t=1}^{T} S_{t,T}, \quad \texttt{Micro} = \frac{\sum_t N_t S_{t,T}}{\sum_t N_t}$$

These metrics provide complementary insights, especially under task imbalance. To quantify forgetting, we use *Average Forgetting*:

$$\texttt{AF} = \frac{1}{T-1} \sum_{t=1}^{T-1} (S_t^{\max} - S_{t,T}),$$

where $S_t^{\max} = \max_{j<T} S_{t,j}$ is peak accuracy for task $t$ before stage $T$. For routing methods, we report task-ID accuracy $\text{Acc}_{\text{task}} = \frac{1}{N} \sum_{i=1}^{N} \mathbb{1}[\hat{t}_i = t_i]$, where $\hat{t}_i$ is the router's predicted task (maximum weight). Calibration is measured via ECE. We partition confidences into $M$ bins and compute:

$$\texttt{ECE} = \sum_{m=1}^{M} \frac{|B_m|}{N} |\text{accuracy}_m - \text{confidence}_m|,$$

where $\text{accuracy}_m$ and $\text{confidence}_m$ are the mean VQA score and confidence of samples that fall into bin $m$ (more details in App. E).

## 5.2. Main Results: Continual VQA Performance

Table 2 summarizes continual VQA results on VQA-v2 and TDIUC under two protocols: question-shift learning (CL-LS) and image-distribution-shift learning (CL-VS). We report Micro and Macro accuracy (higher is better) together with average forgetting (lower is better). See Table 7 in Appendix I for FLAVA results.

Across backbones and datasets, our Bayesian MoE approach achieves the strongest or most competitive accuracy while substantially reducing forgetting. With ViLT on VQA-v2, our method attains **64.16** Micro and **55.82** Macro accuracy with only **0.63** forgetting, outperforming both rehearsal-free baselines (EWC) and replay-based methods (ER/DER/GEM). On TDIUC-LS, we reach **78.81** Micro / **68.48** Macro with **0.40** forgetting, demonstrating consistent retention of earlier tasks. Under the CL-VS distribution shift on TDIUC, our method still improves accuracy over replay baselines (83.41 Micro / 83.74 Macro) while reducing forgetting compared to ER/DER/GEM, although a gap remains to the oracle task-id routing upper baseline.

A notable observation is that our approach matches or even exceeds *Oracle-MoE* routing's performance on VQA-v2, despite oracle's access to the ground-truth task identity. This supports our key premise for continual VQA: due to task overlap, non-native experts can be more competent on a subset of queries, and utility-driven routing with calibrated aggregation can exploit this cross-task competence more effectively than hard task-id selection.

## 5.3. Confidence Calibration Comparison

Figure 6 compares calibration (ECE; lower is better) and accuracy. To isolate calibration effects from performance differences,

*Table 2.* Performance comparison on CL scenarios based on the VQA-v2 dataset and TDIUC dataset with different protocols (CL-LS: Class-Incremental Learning with Label Shift, CL-VS: Class-Incremental Learning with Image distribution Shift).

| Methods | #Mem | VQA-v2 | | | TDIUC | | | | | |
|---|---|---|---|---|---|---|---|---|---|---|
| | | CL-LS | | | CL-LS | | | CL-VS | | |
| | | Micro Acc ↑ | Macro Acc ↑ | Forget ↓ | Micro Acc ↑ | Macro Acc ↑ | Forget ↓ | Micro Acc ↑ | Macro Acc ↑ | Forget ↓ |
| Naive | - | 52.52 | 47.92 | 11.12 | 63.19 | 45.91 | 28.46 | 76.39 | 77.33 | 10.16 |
| EWC | - | 51.39 | 47.31 | 7.06 | 46.01 | 33.24 | 41.03 | 75.08 | 75.82 | 9.71 |
| InfLoRA | - | 49.37 | 45.23 | 13.29 | 52.95 | 39.05 | 25.38 | 80.53 | 81.17 | 6.65 |
| DualPrompt | - | 29.97 | 23.42 | 7.54 | 40.28 | 35.15 | 7.59 | 79.10 | 78.56 | 0.95 |
| ER | 5000 | 58.42 | 53.60 | 3.62 | 67.87 | 61.37 | 5.74 | 79.21 | 79.64 | 5.65 |
| DER | 5000 | 57.92 | 52.91 | 3.12 | 73.40 | 66.20 | 3.87 | 81.18 | 81.62 | 4.06 |
| GEM | 5000 | 54.29 | 49.11 | 7.83 | 75.71 | 66.03 | 2.14 | 81.83 | 82.26 | 4.03 |
| VQACL | 5000 | 41.01 | 36.52 | 1.66 | 53.87 | 45.94 | 7.13 | 67.83 | 68.68 | 10.61 |
| Oracle-MoE | - | 62.20 | 50.84 | 0.00 | 79.57 | 68.70 | 0.00 | 86.96 | 87.04 | 0.00 |
| **Ours** | **5000** | **64.16** | **55.82** | **0.63** | **78.81** | **68.48** | **0.40** | **83.41** | **83.74** | **3.21** |

*Table 3.* Task-wise accuracy, ECE, and task identification (TID) accuracy (only applicable to the Standard method).

**(a) VQA-V2 CL-LS**

| Method | | Rec | Loc | Jud | Com | Cnt | Act | Col | Typ | Sub | Cau |
|---|---|---|---|---|---|---|---|---|---|---|---|
| Oracle | Acc | 50.5 | 21.9 | 81.4 | 76.3 | 49.5 | 63.2 | 81.7 | 26.9 | 45.1 | 11.9 |
| | ECE | .09 | .06 | .19 | .11 | .11 | .16 | .08 | .07 | .05 | .04 |
| Standard | Acc | 50.5 | 29.1 | 81.2 | 77.5 | 49.6 | 72.0 | 81.4 | 54.2 | 52.2 | 12.0 |
| | ECE | .09 | .09 | .19 | .13 | .11 | .18 | .08 | .05 | .12 | .42 |
| | TID | 95 | 60 | 90 | 87 | 96 | 51 | 89 | 0 | 7 | 13 |
| Bayesian | Acc | 50.5 | 30.6 | 81.2 | 77.0 | 49.6 | 72.5 | 81.2 | 54.5 | 52.3 | 13.2 |
| | ECE | .09 | .05 | .19 | .10 | .08 | .12 | .05 | .03 | .06 | .29 |

**(b) TDIUC CL-VS**

| Method | | Ani | Foo | InA | OutA | Tra |
|---|---|---|---|---|---|---|
| Oracle | Acc | 88.5 | 86.4 | 84.4 | 89.7 | 86.2 |
| | ECE | .09 | .11 | .12 | .08 | .10 |
| Standard | Acc | 88.5 | 83.3 | 82.9 | 89.0 | 85.2 |
| | ECE | .09 | .13 | .13 | .08 | .11 |
| | TID | 43 | 38 | 17 | 10 | 84 |
| Bayesian | Acc | 88.5 | 83.8 | 83.5 | 89.6 | 85.9 |
| | ECE | .09 | .09 | .06 | .03 | .05 |

**(c) TDIUC CL-LS**

| Method | | CR | SR | OR | CT | PR |
|---|---|---|---|---|---|---|
| Oracle | Acc | 75.8 | 94.2 | 88.1 | 54.3 | 31.0 |
| | ECE | .22 | .05 | .09 | .37 | .04 |
| Standard | Acc | 75.8 | 93.9 | 87.7 | 54.3 | 32.2 |
| | ECE | .22 | .05 | .10 | .37 | .07 |
| | TID | 99 | 99 | 97 | 100 | 91 |
| Bayesian | Acc | 75.8 | 93.9 | 87.7 | 54.4 | 32.2 |
| | ECE | .22 | .05 | .10 | .37 | .05 |

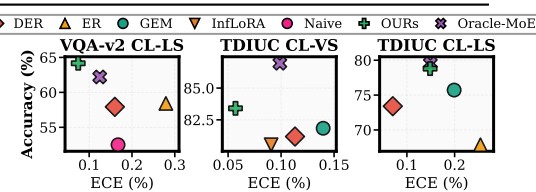

*Figure 6.* Accuracy–ECE comparison among methods with accuracies within ∼10% of ours (lower ECE is better).

we report ECE comparisons only against methods whose accuracy is within ∼ 10% of ours on each benchmark (i.e., matched-accuracy baselines). Under this accuracy-matched regime, our method consistently lies on the top-left region—achieving the lowest ECE while also attaining the highest accuracy among the continual-learning baselines on VQA-v2 and TDIUC CL-VS; the main exception is TDIUC-LS, where DER exhibits lower ECE, but without consistent behavior across datasets and with a less favorable accuracy–calibration trade-off overall.

Notably, compared with Oracle-Moe (ground-truth expert selection) our method matches its ECE on TDIUC-LS, and improves ECE on TDIUC CL-VS and VQA-v2. This pattern is consistent with the role of task overlap and meaningful routing uncertainty: when multiple experts are plausible (coarse visual hyper-categories in TDIUC CL-VS; noisy lexical task boundaries and near-universal answer sharing in VQA-v2), router uncertainty becomes informative, entropy regularization discourages premature one-hot routing, and Bayesian aggregation exploits expert disagreement, reducing overconfidence and improving end-to-end calibration without sacrificing accuracy; when tasks are largely separable (TDIUC-LS), oracle routing already leaves limited headroom for further calibration gains.

## 6. Ablation Studies

### 6.1. Routing and Aggregation: Task-wise Analysis

To unpack *why* our method helps in continual VQA, Tables 3a–c report per-task **Acc/ECE**. for STANDARD (hard routing to top-1 expert) we additionally show **TID**, the agreement between the router's top-1 expert and the benchmark task label. We compare hard routing (STANDARD), oracle task-ID routing (ORACLE), and our Bayesian multi-expert aggregation (BAYESIAN).

**Task-ID accuracy is a weak proxy for VQA accuracy.** STANDARD frequently outperforms ORACLE even when TID is low, indicating beneficial routing to *non-native* experts. On VQA-v2, TYPE has TID = 0 yet attains 54.2% vs. 26.9% by ORACLE; ACTION achieves 72.0% vs. 63.2% (TID = 51); SUBCATEGORY achieves 52.2% vs. 45.1% (TID = 7). This supports framing routing as *utility maximization* rather than task identification.

**Miscalibration can arise from misrouting.** When task boundaries are ambiguous, selecting a single expert amplifies miscalibration. The clearest example is VQA-v2 CAUSAL: STANDARD and ORACLE have similar accuracy (12.0 vs. 11.9), but vastly different calibration (ECE: 0.42 vs. 0.04). Consistently higher ECE under standard routing (vs. oracle) supports our motivation: routing to non-native experts may improve accuracy yet worsen calibration.

Bayesian aggregation mitigates miscalibration by combining evidence from multiple plausible experts, reducing ECE across overlap-heavy tasks (e.g., ACTION 0.18→0.12, SUBCATEGORY 0.12→0.06, CAUSAL 0.42→0.29) while matching or slightly improving accuracy. On TDIUC CL-VS, where categories share visual content and answer space, Bayesian aggregation yields the largest calibration benefits and can even surpass oracle calibration (e.g., OUTA ECE 0.08→0.03; TRA 0.10→0.05). On TDIUC-LS, overlap is weaker, so calibration gains are smaller, but non-degrading, most notably for POSITIONAL REASONING (ECE 0.07→0.05). Overall, aggregation is most effective when task overlap makes router uncertainty meaningful and exploitable.

### 6.2. Component Analysis

We ablate: *utility-trained routing* (UTR), *router entropy regularization* (ER), and *Bayesian aggregation* (BA) at inference. All variants share the same experts and continual training pipeline; only the router objective (task-ID vs. utility) and the inference rule (top-1 vs. aggregation) change. Table 4 compares accuracy, ECE,

*Table 4.* Ablation study on component effectiveness. Config: baseline (—), Bayesian Agg. (B), Utility Router (U), +Entropy Reg. (E). Metrics: Ac/EC/RE/AF/TI = Accuracy/ECE/Router Entropy/Avg. Forgetting/Task-ID Acc.

| | VQA-v2 | | | | | TDIUC-LS | | | | | TDIUC-VS | | | | |
|---|---|---|---|---|---|---|---|---|---|---|---|---|---|---|---|
| Cfg | Ac | EC | RE | AF | TI | Ac | EC | RE | AF | TI | Ac | EC | RE | AF | TI |
| — | 57.2 | .15 | .84 | 3.0 | 72 | 77.7 | .15 | .11 | 1.5 | 96 | 80.4 | .15 | 1.2 | 6.2 | 36 |
| B | 60.3 | .10 | .84 | 1.8 | - | 78.8 | **.13** | .11 | .5 | - | 82.0 | .07 | 1.2 | 5.0 | - |
| U | 63.3 | .15 | .03 | .5 | 63 | 78.5 | .16 | .01 | .7 | 98 | 80.4 | .15 | .06 | 6.6 | 33 |
| U+B | 63.6 | .14 | .03 | **.4** | - | 78.7 | .15 | .01 | .6 | - | 80.5 | .14 | .06 | 6.6 | - |
| U+E | 63.9 | .14 | .93 | .8 | 77 | 78.6 | .16 | .05 | .6 | 98 | 81.6 | .14 | 1.4 | 4.7 | 39 |
| U+E+B | **64.2** | **.07** | **.93** | .6 | - | **78.8** | .15 | **.05** | **.4** | - | **83.4** | **.06** | **1.4** | **3.2** | - |

average forgetting (AF) (TID; reported for hard-routing variants). Baseline router is trained with task-ID supervision.

Comparing **Hard routing** vs. **BA** under task-ID supervised router (rows 1-2), BA consistently reduces ECE and often improves accuracy: On VQA-v2, BA reduces ECE from $0.15 \to 0.10$, while also improving accuracy (e.g., $57.2 \to 60.3$. Similar trends appear under TDIUC-VS, where BA yields large calibration gains $(0.15 \to 0.07)$. This supports the interpretation that Bayesian aggregation reduces overconfident errors primarily through *evidence pooling* across experts and robustness to routing uncertainty.

Comparing **Hard** vs. **UTR-only** (rows 1 and 3), utility supervision substantially improves accuracy and reduces forgetting on VQA-v2 $(57.2 \to 63.25;$ AF $3.0 \to 0.5)$, consistent with our motivation that task-ID routing is suboptimal under task overlap (Fig. 2a, 3). However, UTR dramatically reduces router entropy (RE $0.84 \to 0.03$ on VQA-v2; $0.11 \to 0.01$ on TDIUC-LS), indicating near one-hot routing. This over-sharp behavior also explains why calibration does not improve under UTR alone (ECE is similar or worse than the baseline). In other words, UTR selects more competent experts (higher accuracy) but tends to be overconfident about that selection (low RE), which limits reliability gains. *UTR+ER* increases RE substantially (e.g., on VQA-v2 RE $0.03 \to 0.93$; $0.06 \to 1.4$ on TDIUC-VS), restoring meaningful uncertainty over multiple experts. Yet without BA (i.e. hard routing), these entropy gains alone do not consistently translate into lower ECE—highlighting that ER primarily serves to *enable* effective multi-expert aggregation.

Finally, the **full model** (UTR+ER+BA) achieves the strongest overall performance: it retains the accuracy/forgetting gains of UTR while obtaining the largest ECE reductions via calibrated aggregation. For instance, on VQA-v2 the full model improves accuracy to 64.2 while cutting ECE to 0.07. On TDIUC-VS, it reaches the lowest ECE (0.06) together with the highest accuracy of 83.4. On TDIUC-LS, where task separability is higher (TID 98%), calibration gains are smaller, but the full model maintains comparable ECE, while improving or matching accuracy and forgetting.
This ablation supports three key conclusions: (1) **UTR** is the main driver of accuracy/forgetting improvements by routing to competent (sometimes non-native) experts; (2) **UTR alone** tends to produce over-sharp routing (low RE), limiting calibration; and (3) **ER + BA** is crucial to translate multi-expert knowledge sharing into *calibrated* VQA probabilities, yielding the best end-to-end reliability without sacrificing performance.

### 6.3. Router training and routing mechanisms

Table 5 isolates the effect of routing while holding the expert pool and training stream fixed. As a strong upper baseline, *Oracle Task-ID* hard-routes to the ground-truth task expert, yielding 62.20 ACC with zero forgetting and 100% task-ID agreement. However, this strategy is not utility-optimal in continual VQA: because tasks overlap in both input and answer space, the task's "native" expert is not always the most competent for a given query. Consistent

*Table 5.* Ablation on routing mechanisms and router objectives: average VQA accuracy (ACC), average forgetting (AF), expected calibration error (ECE), and task-ID prediction accuracy ($Acc_{task}$).

| Routing / Router objective | Ac ↑ | AF ↓ | ECE ↓ | $Acc_{task}$ |
|---|---|---|---|---|
| Oracle Task-ID (hard) | 62.20 | 0 | 0.12 | 100.0 |
| Task-ID supervised router (hard) | 57.22 | 3.00 | 0.15 | 72.47 |
| AE-min error router (hard) | 56.72 | 2.99 | 0.12 | 76.95 |
| Utility router + entropy reg | 63.92 | 0.79 | 0.13 | 76.91 |
| Utility router + entropy reg + BA | **64.16** | **0.63** | **0.07** | – |

with this, our *utility-trained router* exceeds oracle performance (63.92 vs. 62.20 ACC) despite lower task-ID agreement (76.91%), indicating that it deliberately routes some examples to non-native experts that answer better. In contrast, both *task-ID supervised* and *AE-min error* routing underperform oracle, with substantially lower ACC (57.22/56.72) and higher forgetting (3.0/2.99). Finally, combining the *entropy-regularized* utility router with *Bayesian aggregation* yields best overall trade-off: it preserves accuracy gains from utility routing (64.16 ACC) while dramatically improving reliability (ECE $0.12 \to 0.07$), supporting our central claim that calibrated, soft routing is crucial for aggregation: entropy regularization maintains non-trivial mass over multiple plausible experts, and Bayesian aggregation pools evidence when experts agree while reducing overconfident errors when they disagree.

### 6.4. Ablation on Memory-buffer Size

Figure 7 varies the replay memory size $|\mathcal{B}|$ between 0–10,000. Without replay ($|\mathcal{B}|=0$), accuracy and calibration degrade noticeably, consistent with the router *catastrophically forgetting* expert competence and producing unreliable routing weights. Once replay is enabled, the method is largely *insensitive* to $|\mathcal{B}|$: increasing the buffer from 200 to 10,000 changes accuracy and ECE only marginally, and even $|\mathcal{B}|=200$ suffices. This is expected because the buffer is used only to rehearse *expert competence for routing*, rather than to rehearse the full VQA model or its output distribution.

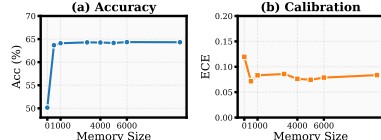

*Figure 7.* Accuracy and ECE are largely insensitive to buffer size.

## 7. Conclusion

Continual VQA violates the usual assumption of cleanly separable tasks: images, question patterns, and answer vocabularies overlap, so task identity is an unreliable proxy for which expert will answer best. In this regime, hard routing wastes transferable knowledge and can become brittle and overconfident when the selected expert is only a partial match to the query. We introduce a calibrated Bayesian Mixture-of-Experts framework that grows parameter-efficient per-task experts, trains a router by directly maximizing expected downstream VQA utility, and marginalizes expert identity via Bayesian aggregation in a unified answer space. To keep aggregation effective, we prevent routing from collapsing toward one-hot decisions by entropy regularization, which preserves meaningful uncertainty and enables evidence pooling across plausible experts. Empirically, this combination improves both continual performance and reliability: on VQA-v2, we outperform the best CL baseline by **+5.74** accuracy while reducing forgetting by **2.99**, and simultaneously cut ECE from **0.15** to **0.07**. These results suggest a practical design principle for continual VQA: train routing for *competence* (not task labels), allow the router to express uncertainty when tasks overlap, and aggregate predictions so that overlap becomes a source of robustness and better-calibrated confidence rather than a failure mode of single-expert selection.

## Acknowledgments

This research was sponsored by the Army Research Office under Grant Number W911NF-24-1-0385. The views and conclusions contained in this document are those of the authors and should not be interpreted as representing the official policies, either expressed or implied, of the Army Research Office or the U.S. Government. The U.S. Government is authorized to reproduce and distribute reprints for Government purposes notwithstanding any copyright notation herein.

## Impact Statement

Continual visual question answering (VQA) is increasingly relevant for deployed vision–language systems that must adapt as visual domains, question patterns, and answer vocabularies evolve in time. In such settings, it is not enough to maintain accuracy: models must also avoid *overconfident* mistakes and provide confidence estimates that are meaningful under shift. In real applications (e.g., assistive interfaces or decision-support workflows), overconfident wrong answers can be severely harmful. Our work advances continual VQA toward this goal by reducing catastrophic forgetting while substantially reducing miscalibration, making VQA predictions more reliable when task boundaries are ambiguous.

From a resource and privacy perspective, continual learning can reduce the compute and energy costs of repeatedly retraining large vision–language models, supporting sustainability goals and green AI endeavors. Our approach is also compatible with lightweight updates and compact replay, which can lower storage requirements and help limit long-term retention of sensitive data. As with all VQA systems, societal risks remain: biases in pretraining data may persist, and better calibration does not guarantee correctness under extreme distribution shifts. As such, high-stakes use should include domain-specific evaluation and appropriate human oversight.

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

# Supplementary Material

## A. Organization of Appendix

In the Appendix, we first summarize the major notations used throughout the paper in Appendix B, and then provide additional related work and detailed descriptions of the compared baselines in Appendices C and D. We next define all evaluation metrics (including ECE and forgetting) in Appendix E, and describe the datasets, continual splits, and overlap regimes analyzed in our experiments (e.g., VQA-v2, TDIUC-VS, and TDIUC-LS) in Appendix F. Further implementation details and extended results are presented in Appendices G-K. Finally, we discuss limitations/future works in L, and provide the anonymous source-code link in Appendix M.

## B. Summary of Notations

Table 6 summarizes the major notations used in our paper.

| Symbol group | Notation | Description |
|---|---|---|
| Data / tasks | $\mathcal{V}, \mathcal{Q}$ | Image space and question space. |
| | $v, q$; $\mathbf{x} = (v, q)$ | Image, question, and their paired VQA input. |
| | $a$ | Candidate answer (class label). |
| | $\mathcal{A}(\mathbf{x})$ | Ground-truth VQA annotations for $x$ (e.g., multiple human answers). |
| | $\mathcal{Y}$ | Unified answer vocabulary. ($\mathcal{Y} = \bigcup_{t=1} \mathcal{Y}_t$: unified answer space). |
| | $\mathcal{Y}_t$ | Task-$t$ local answer set. |
| | $\mathcal{D} = \{(v_n, q_n, a_n)\}_{n=1}^N$ | Dataset with $N$ examples; ($n$: sample index). |
| | $\mathcal{D}_t$ | Dataset for task $\mathcal{T}_t$. |
| | $N_t$ | Number of samples in task $t$. |
| | $\mathcal{T}_t, t, T$ | The $t$-th task in a stream of $T$ tasks. |
| | $s$ | Current continual stage (number of tasks/experts available so far). |
| | $\mathcal{B}, |\mathcal{B}|$ | Replay buffer for router training; $|\mathcal{B}|$ is total buffer size. |
| Models / parameters | $f_\Theta; \Theta$ | Generic VQA model/function and its parameters. |
| | $F_{\theta_0}; \theta_0$ | Frozen pretrained backbone and its parameters. |
| | $E_t(\cdot) / f_t(\cdot)$ | Expert $t$ (backbone + task-specific modules). |
| | $\Delta\theta_t, H_t$ | Task-$t$ LoRA adapter parameters and task-local answer head. |
| | $\Theta^{(t)}$ | Expert-$t$ parameters ($\Theta^{(t)} \equiv (\Delta\theta_t, H_t)$). |
| | $h_t(\mathbf{x}), \ell_t(\mathbf{x})$ | Expert-$t$ representation $h_t(\mathbf{x}) = F_{\theta_0, \Delta\theta_t}(\mathbf{x})$ and task-local logits $\ell_t(\mathbf{x}) = H_t(h_t(\mathbf{x})) \in \mathbb{R}^{|\mathcal{Y}_t|}$. |
| | $r_\varphi(\mathbf{x}), w_\varphi(t \mid x); \varphi$ | Router logits $r_\varphi(\mathbf{x}) \in \mathbb{R}^s$, router parameters $\varphi$, and routing weights $w_\varphi(\cdot \mid \mathbf{x}) = \text{Softmax}(r_\varphi(\mathbf{x})) \in \Delta^{s-1}$. |
| | $W' = W + BA; A, B; r; L$ | LoRA update of a frozen weight $W$ via low-rank factors. $r$ is LoRA rank. $L$ is the number of backbone layers with injected LoRA modules. |
| Probabilities / inference | $p_\Theta(a \mid \mathbf{x})$ | Predictive distribution over answers for input $x$. |
| | $p_t^{\text{loc}}(a \mid \mathbf{x})$ | Expert-$t$ distribution in its local answer space $\mathcal{Y}_t$. |
| | $M_t, p_t(a \mid \mathbf{x})$ | Alignment map/matrix $M_t$ from $\mathcal{Y}_t$ to $\mathcal{Y}$ and unified expert distribution $p_t(\cdot \mid \mathbf{x}) = M_t \, p_t^{\text{loc}}(\cdot \mid \mathbf{x})$. |
| | $z$ | Latent expert index in Bayesian model averaging. |
| | $p(a \mid \mathbf{x})$ | Bayesian aggregation: $p(a \mid \mathbf{x}) = \sum_{t=1}^s w_\varphi(t \mid \mathbf{x}) \, p_t(a \mid \mathbf{x})$. |
| | $\hat{a}(\mathbf{x}), c(\mathbf{x})$ | Prediction and confidence: $\hat{a}(\mathbf{x}) = \arg\max_a p(a \mid \mathbf{x})$, $c(\mathbf{x}) = p(\hat{a}(\mathbf{x}) \mid \mathbf{x})$. |
| | $s_\mathbf{x}(a), U_t(\mathbf{x})$ | VQA score of answer $a$ for example $\mathbf{x}$ (in $[0, 1]$ under standard VQA scoring) and expert utility $U_t(\mathbf{x}) = \mathbb{E}_{a \sim p_t}[s_\mathbf{x}(a)]$. |
| | $H(w), \lambda; k, \mathcal{S}_k(\mathbf{x}), \tilde{w}_\varphi, p_k$ | Entropy $H(w) = -\sum_t w_t \log w_t$ and its coefficient $\lambda$. Top-$k$ aggregation uses the index set $\mathcal{S}_k(\mathbf{x})$, renormalized weights $\tilde{w}_\varphi$, and distribution $p_k(a \mid x)$. |
| Metrics | $S_{i,j}$ | Accuracy on task $i$ after learning stage $j$. |
| | $S_{t,T}, S_t^{\max}$ | Final and peak accuracies for task $t$: $S_t^{\max} = \max_{j<T} S_{t,j}$. |
| | Micro, Macro | Micro (sample-weighted) and Macro (task-average) accuracies. |
| | Forget / AF | Average forgetting: $\frac{1}{T-1} \sum_{t=1}^{T-1}(S_t^{\max} - S_{t,T})$. |
| | ECE; $B_m; M$ | Expected Calibration Error using confidence bins $B_m$ and number of bins $M$. |
| | $\text{Acc}_{\text{task}} / \text{TID}; \mathbb{1}[\cdot]$ | Task-ID accuracy for routing baselines: $\frac{1}{N} \sum_{n=1}^N \mathbb{1}[\hat{t}_n = t_n]$. |

*Table 6.* Summary of notations used in the paper.

## C. Additional Related Work

**Visual question answering.** VQA has evolved rapidly from task-specific architectures toward large-scale vision–language pretraining. Early deep VQA models established the effectiveness of attention for localizing salient image regions and question tokens, together with improved multimodal fusion operators for combining visual and linguistic features (Anderson et al., 2018; Schwartz et al., 2017; Lu et al., 2018; Yu et al., 2019; Gao et al., 2019; 2016; Fukui et al., 2016; Kim et al., 2016; Ben-Younes et al., 2017). More recently, cross-modal Transformer pretraining has become the dominant paradigm, with both two-stream (e.g., ViLBERT, LXMERT) and single-stream (e.g., UNITER) encoders learning aligned representations from large image–text corpora (Lu et al., 2019; Tan & Bansal, 2019; Chen et al.,

2020). Several strong variants further improve alignment by leveraging object-tag signals (OSCAR, VinVL) or by simplifying the pipeline to avoid region-feature extraction altogether (ViLT), enabling efficient end-to-end VQA models (Li et al., 2020; Zhang et al., 2021; Kim et al., 2021). Unified pretraining objectives that mix contrastive, matching, and masked modeling losses (e.g., ALBEF, FLAVA) continue to raise performance and transferability across VQA benchmarks (Li et al., 2021a; Singh et al., 2022). In parallel, large prompted or instruction-tuned VLMs also achieve strong VQA results, though they introduce additional considerations around generation, verbosity, and calibration (Alayrac et al., 2022; Li et al., 2023; Liu et al., 2023). Despite these advances, VQA remains sensitive to language priors and shortcut learning, motivating bias-reduction methods and diagnostic benchmarks (e.g., VQA-CP) (Agrawal et al., 2018; Cao & Li, 2023; Lao et al., 2021; Li et al., 2021b; Ishmam et al., 2025; Zhang et al., 2023a). Reliability has also become a central concern: VQA models can be miscalibrated under distribution shift, motivating selective prediction/abstention and calibration-focused evaluation (Whitehead et al., 2022; Guo et al., 2017; Ovadia et al., 2019).

**Continual learning.** CL studies how to learn from a task stream without catastrophic forgetting; broad strategies include replay, regularization, and architectural isolation/expansion (Wang et al., 2024; Kirkpatrick et al., 2017; Li & Hoiem, 2017; Buzzega et al., 2020). For large pretrained backbones, *parameter-efficient* CL (adapters, prompts, LoRA) has become a practical default, since it reduces interference by constraining updates to small task-specific modules while preserving foundation-model priors. Within this paradigm, *modular and conditional computation*—notably mixture-of-experts (MoE)—is especially relevant: routing/gating decides which subset of modules to activate and update, enabling both task isolation and controlled reuse. Classic routing-based lifelong systems include expert-selection/gating and path/routing policies over modules (Aljundi et al., 2017; Fernando et al., 2017; Shazeer et al., 2017). Recent work has made routing a first-class CL component for pretrained (vision-)language models: MoE-Adapters introduces MoE adapters and a distribution-aware selector that routes in- vs. out-of-distribution inputs to preserve zero-shot behavior (Yu et al., 2024; 2025); and CABLE trains a policy for task-free adapter assignment driven by gradient similarity and performance reward (Julian et al., 2025). Related analyses also study how routers behave under distribution shift during continual pretraining of MoE transformers (Thérien et al., 2025), and very recent work targets *router–expert co-drift* in continual instruction tuning, proposing routing signals tied to expert pathway activations (Hou et al., 2026). These directions closely connect to our setting, where task boundaries are noisy and effective routing must balance specialization with reuse.

**Continual VQA** remains comparatively under-explored, but momentum has accelerated with new benchmarks and MLLM-based continual learners. VQACL formalized a continual VQA setting with dual-level task sequences and compositional generalization tests (Zhang et al., 2023b), while CL-CrossVQA introduced a cross-domain continual VQA benchmark spanning multiple VQA datasets and VL pretrained models (Zhang et al., 2025). On the methods side, prior work spans rehearsal-free prompt learning (e.g., TRIPLET) (Qian et al., 2023), pseudo-rehearsal using VLMs (GaB) (Das et al., 2024), and distillation + replay tailored to multimodal drift (modality-aware feature distillation) (Nikandrou et al., 2024). Recent additions include prompt-based continual VQA (CluMo) (Cai & Rostami, 2025), question-only replay with attention distillation (QUAD) (Marouf et al., 2025), and MoE-based continual VQA with explicit multi-granularity routing (CL-MoE) (Huai et al., 2025); more broadly, continual adaptation of large multimodal models via dual-modality prompting further supports this trend (Li & Lyu, 2025). However, most modular continual VQA approaches still (implicitly or explicitly) treat routing as task identification and/or rely on hard expert choice, which can underutilize cross-task overlap and exacerbate overconfident failures under misrouting.

Our work builds on modular continual VQA, but departs by (i) training the router directly for *expected downstream utility* rather than task-ID supervision, and (ii) *calibrating* multi-expert sharing via Bayesian aggregation and entropy-regularized routing, improving both accuracy/forgetting and probabilistic reliability.

# D. Additional Details on Baselines

**(A) Generic continual-learning baselines.** **(Naive)** Sequentially fine-tune a single model on each task without any forgetting mitigation. **(EWC)** (Kirkpatrick et al., 2017). Regularize updates using parameter importance (Fisher-based) to preserve previous knowledge. **(ER)** (Chaudhry et al., 2018) Maintain an episodic memory and interleave stored samples from past tasks with current-task training. **(GEM)** (Lopez-Paz & Ranzato, 2017) constrains gradient updates during new task learning. **(DER)** (Buzzega et al., 2020). Replay with distillation targets ("dark" outputs / logits) in addition to supervised labels.

**(B) Parameter-efficient CL for pretrained VLMs.** These baselines freeze the pretrained backbone and only learn lightweight, add-on modules (prompts (dualprompt) and low-rank branches (InfLoRA)), making continual adaptation parameter-efficient and reducing interference with the frozen base model. **(DualPrompt)** (Wang et al., 2022a). A prompt-tuning approach that learns two complementary shared *general* prompt (capturing task-invariant knowledge), and *expert*-specific prompts (capturing task-specific knowledge). Prompts are attached to a frozen Transformer backbone; training updates only the prompt parameters (and the task/output head). At inference, expert prompts are selected via a query matching mechanism. **(InfLoRA)** (Liang & Li, 2024). A LoRA-style PEFT method that expands a new low-rank branch per task while keeping the pretrained weights and all previous branches frozen. Crucially, it *pre-designs* the task-$t$ low-rank projection to enforce orthogonality between the update subspace of the new task to gradients of previous tasks, to eliminate inter-task interference while remaining aligned with the new task's gradient subspace. At inference time, all LoRA adapters are used.

**(C) Continual VQA baselines.** VQACL (Zhang et al., 2023b) was originally designed for encoder–decoder VQA architectures (VL-T5), where it learns and retrieves task prototypes and *aggregates* the encoder representation with retrieved prototypes before passing the result through the decoder for answer generation. Since our backbones are *encoder-only* with a classifier head, we adapt VQACL by performing the same prototype retrieval and feature aggregation in the encoder feature space, and then feeding the aggregated representation to the answer classifier (rather than to a decoder). All other components (prototype learning and retrieval) follow the original formulation.

**(D) Modular and MoE routing baselines.** These baselines isolate task-specific parameters but differ in how they *route* inputs to experts. **Task-ID MoE (hard routing).** One expert per task; at test time route each example to exactly one expert using the task ID (oracle). **Task-classifier routing.** Replace oracle task ID with a learned task predictor and route to the predicted expert. Task identity predictor is trained with cross entropy loss, against the actual task identities during training. **Autoencoder routing (AE-min error)** (Aljundi et al., 2017). We train a lightweight autoencoder for each task/expert on that task's training inputs. At test time, given input representation $z(\mathbf{x})$, each autoencoder produces a reconstruction of the corresponding input, and we route to the expert with minimum reconstruction error. $\hat{t}(x) = \arg\min_t \|z(\mathbf{x}) - \tilde{\mathbf{x}}_t(\mathbf{x})\|_2^2$. This provides a fully *task-free* routing mechanism that does not require task-ID supervision and serves as an unsupervised alternative to a task classifier.

## E. Detailed Definitions of Metrics

We report Micro and Macro accuracy (accuracy at sample and task level, respectively), Average Forgetting (AF), and calibration via Expected Calibration Error (ECE) computed from top-1 confidence and VQA scores. For routing analysis we additionally report task-ID agreement ($\mathrm{Acc}_{\mathrm{task}}$) and router entropy when relevant. Let $S_{i,j}$ denote the accuracy of task $i$ at stage $j$. We evaluate accuracy at task, and sample levels, referred to as *Macro Accuracy*, and *Micro Accuracy*:

$$\texttt{Macro Acc} = \frac{1}{T} \sum_{t=1}^{T} S_{t,T} \tag{12}$$

$$\texttt{Micro Acc} = \frac{1}{\sum_t N_t} \sum_{t=1}^{T} N_t S_{t,T} \tag{13}$$

These metrics together provide complementary insights on model performance, giving a fuller picture of model's behavior, especially in scenarios with severe task imbalance. To quantify forgetting of past tasks, we use *Average Forgetting*:

$$\texttt{Forget} = \frac{1}{T-1} \sum_{t=1}^{T-1} S_t^{\max} - S_{t,T}, \tag{14}$$

where $S_t^{\max}$ is the highest accuracy achieved for task $t$ before the final stage.

For routing-based MoE methods, we also report how often the router selects the benchmark-provided task label. Let $t_i$ be the ground-truth task ID for example $\mathbf{x}_i$, and let the router output either a distribution $p_\phi(t \mid x_i)$ or a hard selection. We define the predicted task as $\hat{t}_i = \arg\max_t p_\phi(t \mid x_i)$ (or $\hat{t}_i$ given directly by the routing rule, e.g., AE-min error), and report

$$\mathrm{Acc}_{\mathrm{task}} = \frac{1}{N} \sum_{i=1}^{N} \mathbb{1}\left[\hat{t}_i = t_i\right].$$

We evaluate the calibration performance by ECE. Given $N$ evaluated examples with confidences $\{c_i\}_{i=1}^N$ and VQA scores $\{s_i\}_{i=1}^N$, we partition the confidence interval $[0, 1]$ into $M$ bins $\{I_m\}_{m=1}^M$ and define $B_m = \{\, i : c_i \in I_m \,\}$. For each non-empty bin, we compute the empirical accuracy and mean confidence as $\mathrm{acc}(B_m) = \frac{1}{|B_m|} \sum_{i \in B_m} s_i$, and $\mathrm{conf}(B_m) = \frac{1}{|B_m|} \sum_{i \in B_m} c_i$. ECE is then estimated as:

$$\mathrm{ECE} = \sum_{m=1}^{M} \frac{|B_m|}{N} \left| \mathrm{acc}(B_m) - \mathrm{conf}(B_m) \right|.$$

## F. Additional Details on Dataset and Incremental Scenarios

In continual VQA *task boundaries are rarely clean*: tasks can overlap in (i) question semantics, (ii) image content, and (iii) answer vocabularies. Our experimental suite intentionally spans three regimes of overlap. This diversity is important for interpreting both routing behavior and calibration, because Bayesian aggregation can only help when (a) multiple experts are plausible and (b) their predictions can be meaningfully combined in a shared answer space.

**Three overlap regimes (used throughout the paper).** We evaluate continual VQA on two benchmarks (VQA-v2 and TDIUC) under three task streams designed to span distinct degrees of *input* and *output* overlap. **(1) VQA-v2 (question-type tasks):** tasks are induced by question-type heuristics and therefore exhibit semantic overlap and noisy boundaries. Both question patterns and answer vocabularies overlap substantially across tasks. **(2) TDIUC CL-LS (question-type tasks):** tasks are defined by coarse-grained question categories and are comparatively more separable; answer-space overlap is generally small except for the large positional-reasoning task. **(3) TDIUC CL-VS (visual-domain tasks):** tasks are defined by image hyper-categories (dominant visual content), so question types remain broadly similar across tasks and the resulting tasks are highly overlapping in both inputs and outputs.

### F.1. VQA-v2 continual split: question-type tasks with noisy boundaries

We follow the question-type continual learning protocol on VQA-v2 (Goyal et al., 2017) following (Zhang et al., 2023b), consisting of 10 tasks: *recognition, location, judge, commonsense, count, action, color, subcategory, type, causal*. In practice, these partitions are produced

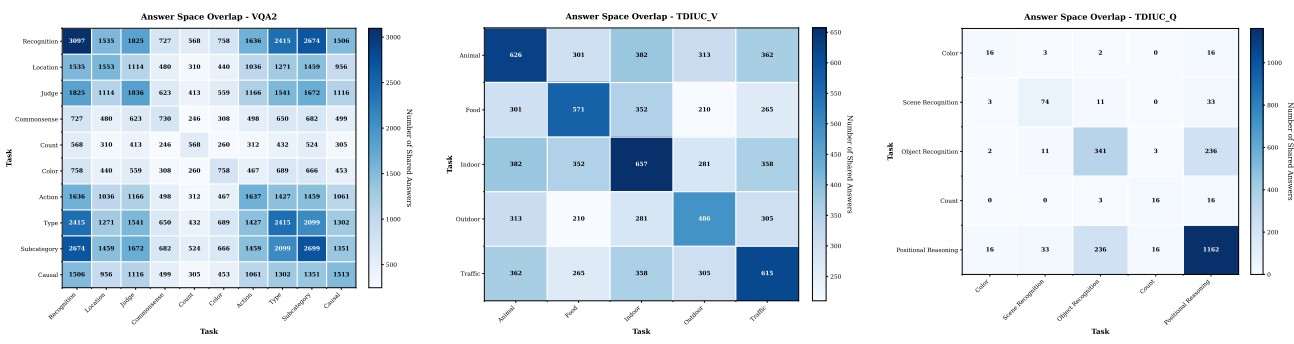

*(a)* VQA-v2 CL-LS (question-type tasks).     *(b)* TDIUC CL-VS (visual-domain tasks).     *(c)* TDIUC CL-LS (question-type tasks).

*Figure 8.* Absolute answer-space overlap matrices $|\mathcal{A}_i \cap \mathcal{A}_j|$. Darker entries indicate larger overlaps.

by lexical/heuristic matching rules on the question text, which induces *noisy task labels* and semantic overlap. For example, questions beginning with "what is the color of ..." are assigned to COLOR, while semantically similar queries such as "what is the main color ..." can fall into RECOGNITION if they do not match the lexical rule. Likewise, multi-skill questions (e.g., color+count) are forced into a single task even when they naturally belong to multiple question types.

This regime stresses test-time routing: knowing the dataset-provided task label does not necessarily identify the expert that will answer the example best, and a router trained to predict task ID can be systematically misled by the partitioning noise. This is precisely the setting where utility-based routing and calibrated multi-expert aggregation are most relevant.

### F.2. TDIUC continual splits: disjoint question-type vs overlapping visual-domain tasking

TDIUC dataset (Kafle & Kanan, 2017) is a widely used benchmark for visual question answering (VQA), built upon images from the Visual Genome (Krishna et al., 2017) and COCO (Lin et al., 2014) datasets. This dataset is specifically designed to facilitate unbiased evaluation of VQA models. We adopt the two continual tasking protocols on TDIUC (Kafle & Kanan, 2017) introduced in prior work (Qian et al., 2023): CL-LS (language shift) and CL-VS (vision shift).

**TDIUC (CL-LS): relatively separable question-type tasks.** In CL-LS, tasks are defined by shifts in question-type distribution. Following (Qian et al., 2023) we use five question-type tasks: *color, scene recognition, object recognition, counting, positional reasoning*. Compared to VQA-v2, these tasks are more cleanly defined (less lexical boundary noise) and tend to be easier to discriminate from question text alone.

**TDIUC (CL-VS): highly overlapping tasks defined by visual content.** In CL-VS, tasks are defined by image hyper-categories (dominant visual content), and we follow the five-way partition: *animal, food, indoor activity, outdoor activity, traffic*. Because each visual-domain task still contains a broad mix of question types, the *input* distributions overlap heavily across tasks, and the output vocabularies remain substantially shared.

### F.3. Answer-space overlap analysis and implications

Let $A_t$ denote the set of unique answers observed in task $t$. Fig. 8 visualizes the absolute overlap matrix $O_{ij} = |A_i \cap A_j|$ for each benchmark.

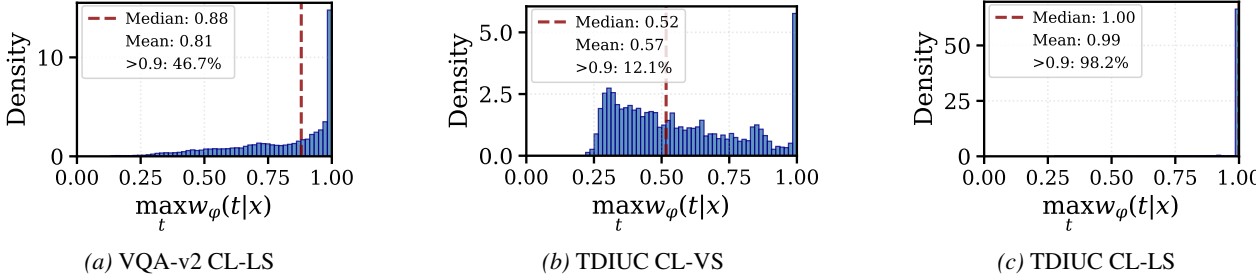

*(a)* VQA-v2 CL-LS     *(b)* TDIUC CL-VS     *(c)* TDIUC CL-LS

*Figure 9.* Confidence distribution of maximum router weight over all tasks, on a) VQA-v2, b) TDIUC CL-VS, c) TDIUC-Q.

**VQA-v2: near-universal answer overlap.** VQA-v2 exhibits extremely high answer sharing across question-type tasks (Fig. 8a). In particular, the RECOGNITION task vocabulary nearly covers the global union of answers, indicating that most tasks share a large

common answer core. Operationally, this makes VQA-v2 well-suited to answer-level aggregation: experts trained on different question types still allocate probability mass to many of the same candidate answers.

**TDIUC (CL-VS): strong answer overlap under visual-domain tasking.** Although TDIUC CL-VS tasks are defined by visual domain, their answer spaces remain heavily shared (Fig. 8b): every task pair shares a large number of answers. Thus, even domain-based tasking does not isolate the output space, and multiple experts can provide meaningful evidence for the same candidate answers.

**TDIUC (CL-LS): small overlaps except for positional reasoning.** TDIUC-Q shows a more separable structure (Fig. 8c). Several smaller tasks have minimal overlap with each other (e.g., COLOR vs. COUNTING), while POSITIONAL REASONING has a much larger answer vocabulary and therefore overlaps with multiple tasks. This asymmetry matters for calibration: overlap-driven gains are expected mainly where multiple experts share answer support (most notably positional reasoning).

### F.4. Interpreting routing, forgetting, and calibration across the three regimes

Our results can be interpreted through the interaction of: (i) task overlap, (ii) router sharpness, and (iii) whether expert predictions can be combined meaningfully.

**TDIUC (CL-VS) (overlap-heavy; easiest for most baselines): largest calibration gains from ER+BA.** Across methods, TDIUC-VS tends to be the least brittle continual setting: tasks share question types and answer vocabularies, so sequential learning induces less destructive interference and even naive baselines can remain reasonably stable. Importantly, this is also the regime where calibrated aggregation helps the most. When tasks overlap, there are often *multiple plausible experts* for a query; entropy regularization prevents the utility objective from collapsing routing to a near one-hot vertex (Fig. 9b), and Bayesian aggregation can then pool evidence on shared answers when experts agree, while preserving uncertainty when they disagree. Even if individual experts are not explicitly calibrated, the mixture reduces overconfident failures by construction, provided the router maintains non-trivial mass on multiple experts.

**TDIUC (CL-LS) (most separable): router becomes sharp; aggregation yields limited calibration gains.** In the relatively disjoint TDIUC-LS setting, a utility-trained router naturally assigns sharp probabilities (Fig. 9c) because there is typically a single clearly best expert. In this case, Bayesian aggregation reduces to (approximately) single-expert prediction, so calibration gains are expected to be small. The main exception is POSITIONAL REASONING, which exhibits the largest answer-space overlap with other tasks (Fig. 8c); here, multiple experts may place mass on shared answers, so aggregation still provide a modest calibration benefit. Specifically our Bayesian aggregation reduces the ECE of POSITIONAL REASONING from 0.07 (of Standard) to 0.05 (See Table 3).

**VQA-v2 (CL-LS) (noisy boundaries; heterogeneous overlap): gains track ambiguity and overlap.** VQA-v2 combines (a) large answer-space overlap with (b) noisy task labels produced by lexical partitioning. This creates systematic ambiguity between certain task pairs (e.g., ACTION vs. JUDGE), where task-ID is not a reliable indicator of the best expert. In such cases, utility-based routing improves accuracy by selecting competent (sometimes non-native) experts, and ER+BA improves reliability by preventing a single misrouted expert from dominating the confidence when task identity is ambiguous. As a consequence, the largest calibration improvements occur on the most overlap-heavy and boundary-ambiguous tasks, while more specialized tasks (with smaller effective overlap) see smaller changes.

## G. Additional Experimental Details and Reproducibility

In this section the hyperparameters used for the experiments and implementation details are presented. All experiments start from pretrained ViLT and FLAVA backbones, with the base encoder kept frozen throughout continual learning; after each task, previously learned experts are frozen as well. We insert LoRA adapters with rank $r = 8$ (and $\alpha = 32$) into all feed-forward (FFN) modules. Each task uses a separate head implemented as a lightweight MLP classifier (Linear $\rightarrow$ LayerNorm $\rightarrow$ GELU $\rightarrow$ Linear) on top of the frozen backbone + expert features. The router is a 2-layer MLP (hidden dim 64, LayerNorm, ReLU, dropout 0.1) operating on question features derived from BERT CLS embeddings. We train the router for 5 epochs and each expert for 10 epochs using AdamW, with learning rate $1e - 3$ with a linear scheduler and warmup. Batch size is 128 for ViLT and 32 for FLAVA. When reporting Bayesian aggregation, we use top-$k$ with $k = 3$, and the entropy-regularized utility-router setting is reported with $\lambda = 0.1$. We conduct all experiments on an NVIDIA RTX A6000 GPU.

## H. Additional Discussion

### H.1. Sensitivity to Task-Boundary Availability

Our approach assumes access to task identifiers (or task boundaries) during *training*. This assumption is standard in much of the continual learning literature, because it enables controlled analysis of forgetting and transfer under a well-defined stream (Zhang et al., 2023b; Abati et al., 2020; Von Oswald et al., 2019; Wortsman et al., 2020). Nevertheless, one may view it as restrictive in deployments where task boundaries are not explicitly provided. Below we clarify why this requirement is typically mild for the VQA settings we study, and why (importantly) it does not conflict with the key motivations and empirical findings of this work.

*Table 7.* Performance comparison on CL scenarios based on the VQA-v2 dataset and TDIUC dataset with different protocols (CL-LS: Class-Incremental Learning with Language distribution Shift, CL-VS: Class-Incremental Learning with Image distribution Shift).

| Model | Methods | #Mem | VQA-v2 | | | TDIUC | | | | | |
|---|---|---|---|---|---|---|---|---|---|---|---|
| | | | CL-LS | | | CL-LS | | | CL-VS | | |
| | | | Micro Acc ↑ | Macro Acc ↑ | Forget ↓ | Micro Acc ↑ | Macro Acc ↑ | Forget ↓ | Micro Acc ↑ | Macro Acc ↑ | Forget ↓ |
| FLAVA | Naive | - | 42.56 | 38.75 | 10.39 | 64.16 | 47.40 | 28.61 | 77.39 | 78.27 | 8.62 |
| | InfLoRA | - | 56.74 | 50.96 | 10.83 | 58.02 | 41.58 | 35.69 | 79.77 | 80.40 | 7.99 |
| | DualPrompt | - | 42.96 | 37.09 | 5.24 | 36.96 | 25.83 | 28.92 | 84.02 | 84.20 | 1.75 |
| | VQACL | 5000 | 51.08 | 47.00 | 6.41 | 40.28 | 37.75 | 8.69 | 64.55 | 65.46 | 8.14 |
| | Oracle-MoE | - | 67.92 | 60.83 | 0.00 | 82.48 | 73.69 | 0.00 | 88.25 | 88.32 | 0.00 |
| | **Ours** | **5000** | **68.29** | **61.25** | **0.50** | **81.86** | **73.12** | **0.56** | **84.15** | **84.46** | **3.73** |

*Table 8.* Ablation study on component effectiveness. U/E/B: Utility Router/Entropy Reg./Bayesian Agg. Ac/EC/RE/AF/TI: Accuracy/ECE/Router Entropy/Avg. Forgetting/Task-ID Acc. **Bold**=best, underline=2nd best.

| Mdl | UTR | ER | BA | VQA-v2 | | | | | TDIUC-LS | | | | | TDIUC-VS | | | | |
|---|---|---|---|---|---|---|---|---|---|---|---|---|---|---|---|---|---|---|
| | | | | Acc | ECE | RE | AF | TID | Acc | ECE | RE | AF | TID | Acc | ECE | RE | AF | TID |
| FLAVA | | | | 65.0 | .16 | .70 | 2.4 | 77 | 81.4 | .15 | .10 | .8 | 96 | 81.6 | .15 | 1.16 | 6.0 | 37 |
| | | | ● | 67.6 | .11 | .70 | **.5** | – | 82.0 | .13 | .10 | **.4** | – | 82.6 | .07 | 1.16 | 5.6 | – |
| | ● | | | 64.1 | .17 | .10 | 1.4 | 59 | 81.5 | .15 | .02 | .9 | 96 | 80.0 | .17 | .10 | 7.2 | 24 |
| | ● | | ● | 64.7 | .15 | .10 | 1.1 | – | 81.6 | .15 | .02 | .9 | - | 80.1 | .15 | .10 | 7.1 | – |
| | ● | ● | | 66.9 | .17 | .83 | 1.1 | 81 | 81.6 | .15 | .10 | .7 | 96 | 82.6 | .14 | 1.4 | 4.7 | 38 |
| | ● | ● | ● | **68.3** | **.09** | .83 | .5 | – | **81.9** | .14 | .10 | .6 | – | **84.2** | **.05** | 1.4 | 3.7 | – |

**Task boundaries need not be clean or perfectly separated.**   Although we use task IDs as supervision during training, the method does *not* rely on tasks being semantically disjoint or sharply separated. In fact, the continual scenarios constructed from VQA-v2 in our experiments are inherently *ill-defined*: tasks are formed using simple lexical rules on question prefixes, which induces substantial semantic overlap across tasks. For instance, questions starting with "what is the color of the" may be mapped to a *color* task, whereas a semantically equivalent query such as "what is the main color" can fail the lexical rule and be assigned to a different task (e.g., *recognition*). Similarly, many questions naturally span multiple skills (e.g., "How many blue circles …"), yet are forced into a single task by the partitioning heuristic. In this regime, task IDs should be interpreted as *weak* supervision: they provide a coarse organizing signal rather than a guarantee of separability.

**Why overlap is not a drawback, but a setting where our method is most useful.**   A core empirical takeaway from our experiments is that *the presence of overlap is precisely where our approach is most beneficial for calibration*. When tasks are semantically entangled, a model that is forced into hard, single-task decisions is more likely to become overconfident on ambiguous inputs. In contrast, our framework can leverage shared structure across tasks: the learned components can effectively treat task supervision as soft guidance and reduce overconfidence on boundary cases, improving calibration. This interpretation is consistent with our observations on TDIUC-LS, where tasks are comparatively more disjoint: because task separability is higher, the opportunity to gain calibration improvements from exploiting cross-task ambiguity is reduced, and we correspondingly observe more limited calibration gains.

**Practicality and extensions beyond explicit task IDs.**   Finally, we emphasize that requiring training-time task IDs is often realistic in practice, even when "true" boundaries are not available: many data streams come with weak metadata that can serve as task tags (e.g., question templates, dataset source, annotation type, or collection regime). Moreover, our setup is compatible with relaxing this assumption by replacing gold task IDs with approximate ones, which would preserve the training recipe while moving toward fully task-agnostic streams. We leave a systematic study of such mechanisms as future work, and note that our current experimental results already operate in regimes where the provided task boundaries are demonstrably noisy and overlapping.

# I. Additional Results on FLAVA

Tables 7–8 report continual VQA results and component ablations for FLAVA. We summarize here the key takeaways for the FLAVA backbone and show that our conclusions are consistent across architectures.

## I.1. Main Results with FLAVA: Trends Match ViLT

Table 7 includes the corresponding FLAVA results under the same continual protocols (VQA-v2 CL-LS, TDIUC CL-LS, and TDIUC CL-VS) and the same evaluation metrics (`Micro`/`Macro` accuracy and `Forget`). Overall, the qualitative picture mirrors the ViLT findings in the main paper:

Across continual streams, our Bayesian MoE remains among the strongest approaches, achieving high (often best) accuracy while substantially reducing forgetting relative to CL baselines (InfLoRA, DualPrompt, and VQACL). In particular, the *forgetting reduction* remains a robust benefit under FLAVA, reinforcing that the improvements are not tied to a specific model choice.

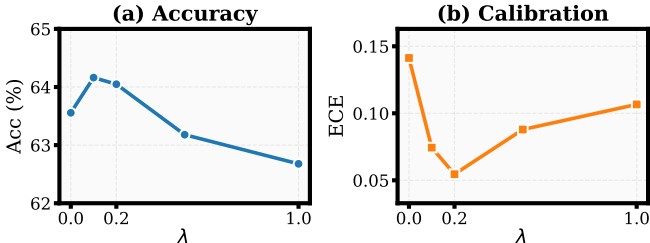

*Figure 10.* Effect of entropy regularization coefficient $\lambda$ on the Accuracy–ECE trade-off, on VQA-v2 (ViLT).

The FLAVA results also follow the regime-dependent behavior described in Appendix F: (i) in overlap-heavy settings (VQA-v2 and especially TDIUC CL-VS), calibrated multi-expert aggregation is most useful because multiple experts remain plausible and can contribute evidence in a shared answer space; (ii) in more separable settings (TDIUC CL-LS), routing naturally becomes sharp and Bayesian aggregation tends to collapse toward single-expert prediction, limiting the headroom for further gains beyond strong routing and stable expert isolation.

Similar to ViLT, the oracle task-ID MoE provides an informative upper baseline on TDIUC CL-VS. While our method narrows this gap via utility-driven routing and calibrated aggregation, the remaining difference is consistent with the intrinsic difficulty of fully task-agnostic routing under heavy visual overlap.

### I.2. Component Ablations with FLAVA

Table 8 further shows that the same components that matter for ViLT remain important under FLAVA:

**Utility routing vs. task-supervised routing.**   Training the router to optimize expected downstream utility consistently improves continual performance over task-ID supervision, aligning with our central motivation that dataset-provided task IDs are an imperfect proxy for the best expert in overlapping continual VQA streams.

**Entropy regularization enables effective Bayesian aggregation.**   Entropy regularization is again critical for preventing degenerate one-hot routing when optimizing a linear utility objective, preserving meaningful router uncertainty that Bayesian aggregation can exploit. Without this, aggregation provides limited benefit because the mixture effectively reduces to hard expert selection.

## J. Additional Ablation Studies

### J.1. Memory-size Sensitivity vs. Rehearsal Baselines

We complement the main memory-size ablation (Fig. 7) with a direct comparison to rehearsal baselines that replay buffered examples through the *VQA model itself*. Table 9 reports Acc/ECE/AF for ER and DER at $\mathcal{B}$=500 and $\mathcal{B}$=5,000, alongside our router-only replay.

For ER/DER, increasing $M$ yields a clear accuracy and forgetting improvement. This is expected: rehearsal methods rely on the buffer to preserve both (i) the evolving *input* distribution (images/questions) and (ii) the *output* behavior over a large, partially overlapping answer space, making performance strongly dependent on buffer capacity. In contrast, our method changes minimally across the same range because replay is *answer-agnostic* and applied only to the router; once the buffer contains a moderately diverse set of past questions and cached expert-utility signals, larger buffers provide limited additional benefit.

*Table 9.* Memory-size sensitivity comparison with replay-based baselines (Acc/ECE/AF format).

| Method | $M = 500$ | $M = 5000$ |
|--------|-----------|------------|
| ER | 54.81 / 0.24 / 7.13 | 58.42 / 0.27 / 3.62 |
| DER | 55.71 / 0.22 / 5.56 | 57.92 / 0.17 / 3.12 |
| Ours | 63.66 / 0.07 / 1.32 | 64.16 / 0.07 / 0.63 |

### J.2. Impact of Entropy Regularization Strength

We study the effect of the entropy regularization coefficient $\lambda$ in Eq. (7), which directly controls the sharpness of the router distribution $w_\varphi(\cdot \mid x)$. Since the utility objective is linear over the simplex, $\lambda \to 0$ naturally drives near one-hot routing; increasing $\lambda$ counteracts this collapse and preserves meaningful uncertainty over multiple plausible experts, which is critical for effective Bayesian aggregation.

Figure 10 shows a clear trade-off curve as we sweep $\lambda \in \{0, 0.1, 0.2, 0.5, 1.0\}$. Moving from $\lambda = 0$ to moderate values ($\lambda \approx 0.1$–$0.2$) improves *both* accuracy and calibration under Bayesian aggregation: Bayesian accuracy increases from 63.56 to $\approx 64.16/64.05$, while Bayesian ECE drops sharply from 0.141 to 0.074/0.055. However, for larger regularization ($\lambda \geq 0.5$), both metrics deteriorate (e.g., Bayesian accuracy drops to $63.18 \to 62.68$ and ECE increases to $0.088 \to 0.107$). We attribute this to **over-smoothing of routing**:

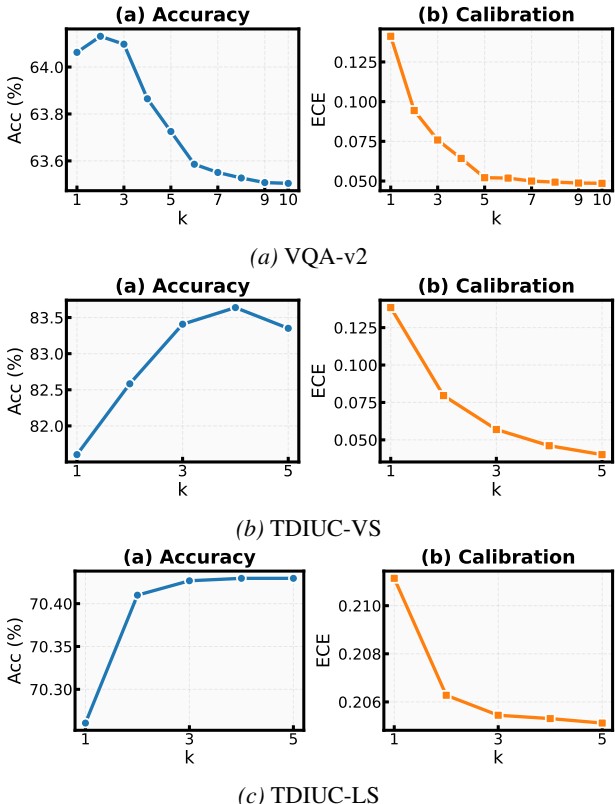

*Figure 11.* Effect of top-$k$ Bayesian Aggregation on the Accuracy–ECE performance, on a) VQA-v2, b) TDIUC-VS, c) TDIUC-LS (ViLT).

when $\lambda$ is too large, the router is encouraged to maintain high entropy even when one expert is clearly superior, forcing probability mass onto weaker experts. This dilutes strong predictions and injects noise into the aggregated posterior, hurting utility (accuracy) and making the confidence–correctness alignment worse (higher ECE). Throughout our experiments, we use $\lambda = 0.1$ as the regularization coefficient.

### J.3. Effect of Number of Aggregated Experts: Top-$k$ Analysis

Figure 11 studies the effect of aggregating the top-$k$ experts at inference time for ViLT model on all three scenarios. When $k = 1$, the model reduces to **hard routing** (no aggregation).

On **VQA-v2** (Figure 11a), accuracy is relatively stable but shows a mild degradation as $k$ grows: it peaks at $k = 3$ (64.13%) and drops from 64.06% at $k = 1$ to 63.50% at $k = 10$ ($\sim 0.56\%$). In contrast, calibration improves substantially and monotonically with $k$, with ECE decreasing from 0.14 ($k = 1$) to 0.05 ($k = 10$). This behavior is consistent with an ensemble-like smoothing effect on the predictive distribution: aggregating more experts reduces overconfidence, but can dilute the strongest expert's prediction on VQA-v2.

On **TDIUC-VS** (Figure 11b), top-$k$ aggregation improves *both* accuracy and calibration. Accuracy increases from 81.60% at $k = 1$ to a maximum of 83.64% at $k = 4$ (and remains high at 83.35% for $k = 5$), while ECE decreases monotonically from 0.14 ($k = 1$) to 0.04 ($k = 5$). The consistent gains suggest that, for TDIUC-VS, multiple high-ranked experts provide complementary evidence such that aggregation improves the final prediction and its calibration rather than injecting noise.

On **TDIUC-LS** (Figure 11c), top-$k$ aggregation has only marginal improvements on both accuracy and calibration, due to the disjoint nature of the tasks within this continual sequence.

Overall, $k = 3$ offers a strong default balance across datasets, while $k = 4$ is best for peak accuracy on TDIUC CL-VS and larger $k$ (up to all experts) may be preferred when well-calibrated confidence is more important than marginal accuracy.

### J.4. Effect of Expert Calibration vs. Calibrated Knowledge Aggregation

Table 10 disentangles two distinct sources of calibration improvement: (i) *per-expert logit calibration* via post-hoc vector scaling (VS), and (ii) *calibrated knowledge aggregation* via a utility-trained router coupled with Bayesian aggregation over experts. We ablate on the effects of the components of our method, and post-hoc calibration, to study the role of each component, in comparison to post-hoc calibration.

*Table 10.* **Expert calibration vs. calibrated knowledge aggregation on VQA-v2.** VS: per-expert vector scaling (held-out validation). "Single expert" selects one expert; "Bayesian agg." combines experts using router probabilities. Entropy regularization (EntReg) is applied only to the utility-trained router.

| Router | VS | Single expert | | Bayesian agg. | |
|---|---|---|---|---|---|
| | | Acc↑ | ECE↓ | Acc↑ | ECE↓ |
| Task-sup. | ✗ | 53.6 | 0.13 | 60.3 | 0.10 |
| | ✓ | 53.3 | 0.11 | 58.5 | 0.08 |
| Utility (no EntReg) | ✗ | 61.7 | 0.16 | 61.9 | 0.16 |
| | ✓ | 61.4 | 0.12 | 61.6 | 0.12 |
| Utility + EntReg | ✗ | **64.1** | 0.14 | **64.1** | 0.07 |
| | ✓ | 63.7 | 0.10 | 63.8 | **0.05** |

Across router types and inference modes, VS consistently reduces ECE (e.g., 0.14→0.10 for utility+EntReg with single-expert inference, and 0.07→0.05 for utility+EntReg with Bayesian aggregation). Notably, the best-calibrated configuration combines both mechanisms: Our utility routing with EntReg and Bayesian aggregation reaches ECE 0.07, and VS further improves to 0.05. This additive improvement supports the interpretation that post-hoc calibration methods such as VS are largely *orthogonal* to our aggregation-driven calibration: VS reshapes each expert's internal confidence distribution (within-expert miscalibration), while Bayesian aggregation primarily reduces overconfidence induced by routing uncertainty and expert disagreement (between-expert uncertainty).

**Accuracy–calibration trade-offs depend on where calibration is applied.** Per-expert calibration can slightly reduce accuracy (e.g., 64.1→63.8 for the full method), consistent with post-hoc scaling occasionally changing the argmax or perturbing mixtures in a way that benefits confidence alignment more than top-1 correctness. Interestingly, this accuracy drop is small for the utility+EntReg+Bayes setting compared to task-supervised Bayes (60.3→58.5), suggesting that a better-trained and regularized router yields a more robust operating point where calibration improvements do not destabilize decisions.

Overall, these results show that well-calibrated confidence is not obtained "for free" by calibrating experts alone: the dominant gains come from *calibrated knowledge aggregation* (utility-trained router with controlled entropy plus Bayesian aggregation), and per-expert calibration provides an additional, complementary improvement.

# K. Qualitative Analysis

## K.1. Confidence Distribution Analysis for Correct and Incorrect Predictions

Figure 12 visualizes the distribution of predicted confidence for correct (left) and incorrect (right) predictions under our utility-trained, entropy-regularized router, comparing hard routing (*Standard*) against Bayesian aggregation (*Bayesian*). Across datasets, hard routing exhibits a clear overconfidence mode on errors: a non-trivial fraction of incorrect predictions concentrate near confidence 1.0 (most prominently on VQA-v2 and TDIUC CL-VS). In contrast, Bayesian aggregation reduces this high-confidence tail for incorrect answers by marginalizing over routing uncertainty, shifting probability mass toward lower confidence values while largely preserving the sharp peak near 1.0 for correct predictions. The effect is weaker on TDIUC CL-LS, where both approaches remain highly confident even when wrong, indicating residual miscalibration not fully explained by routing uncertainty. Overall, Bayesian aggregation improves calibration without sacrificing accuracy on correct examples.

# L. Limitations and Future Work

A key limitation revealed by our analysis is that the calibration gains from Bayesian aggregation are strongly mediated by *task overlap*. When tasks share substantial input structure and especially in output answer space, the router's posterior uncertainty can be informative and aggregation can meaningfully pool evidence; however, in low-overlap or near-separable streams (e.g., TDIUC-LS), the router naturally becomes close to one-hot and aggregation collapses to single-expert behavior, leaving limited headroom for improvement. This also exposes a broader suboptimality in our current design choice: experts are trained in isolation and we rely almost entirely on *inference-time* mixing for knowledge sharing, which limits positive transfer and can under-develop per-expert competence. Future work should therefore explore training-time mechanisms that enable *controlled* sharing—e.g., similarity-aware partial parameter sharing, selective cross-task replay, hierarchical/clustered experts, or interference-constrained joint updates—so that experts become complementary without destructive interference. An intriguing (and currently speculative) direction is to *induce* useful overlap even in nominally disjoint settings—via auxiliary objectives, shared latent answer representations, or other "overlap-imposing" training tricks—so that aggregation remains meaningful rather than degenerating.

A second limitation is that aggregation is ultimately bounded by the reliability of the constituent experts: if individual experts remain overconfident or miscalibrated on non-native inputs, Bayesian averaging may inherit these biases and the per-expert calibration state becomes the performance cap. While our router calibration mitigates overconfident *selection*, we do not directly optimize expert posteriors for calibrated probabilities. Future work will incorporate stronger within-expert calibration (e.g., calibration-aware objectives,

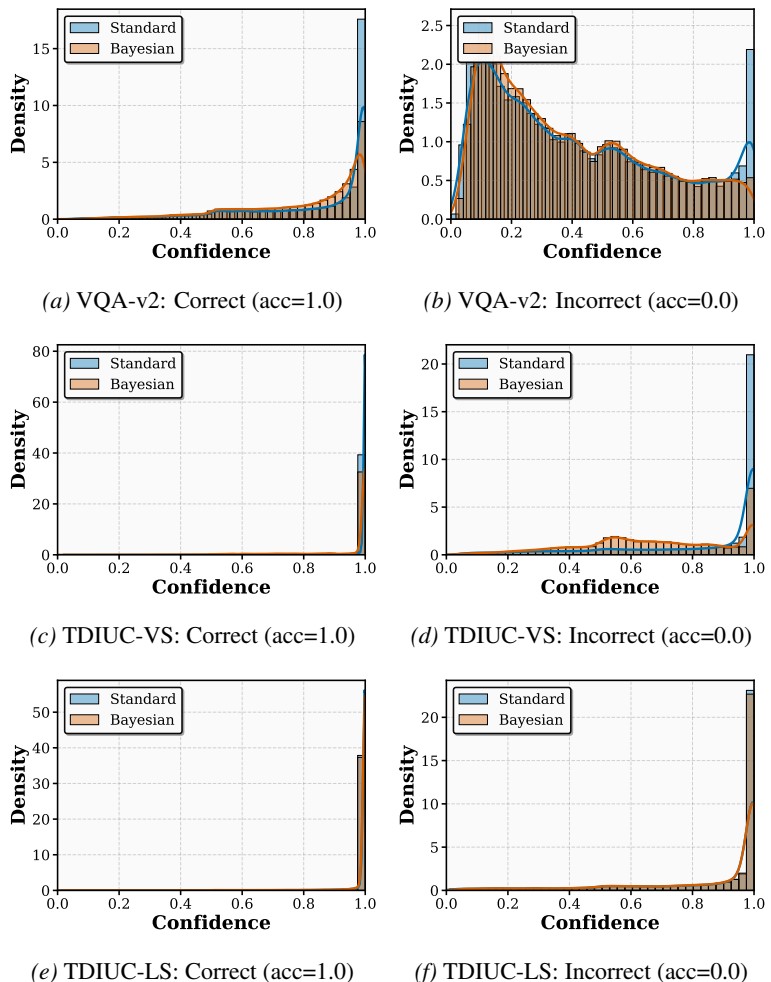

*(a)* VQA-v2: Correct (acc=1.0)    *(b)* VQA-v2: Incorrect (acc=0.0)

*(c)* TDIUC-VS: Correct (acc=1.0)    *(d)* TDIUC-VS: Incorrect (acc=0.0)

*(e)* TDIUC-LS: Correct (acc=1.0)    *(f)* TDIUC-LS: Incorrect (acc=0.0)

*Figure 12.* **Confidence distributions comparing Standard (hard) routing vs Bayesian aggregation, under our utility-trained and entropy regularized router.** Left: correct predictions (acc=1.0). Right: incorrect predictions (acc=0.0). Rows show VQA-V2 (top), TDIUC CL-VS (middle), TDIUC CL-LS (bottom). On VQA-V2 and TDIUC CL-VS, Bayesian aggregation improves calibration by reducing overconfidence on errors while maintaining high confidence on correct predictions.

or uncertainty-aware training) and study joint calibration of router and experts so that "between-expert" uncertainty and "within-expert" confidence are both trustworthy. Finally, our framework assumes access to task boundaries during training to spawn new experts; relaxing this requirement toward task-free expert creation/merging remains an important step for broader applicability.

# M. Source Code

For the source code of this paper, please click here[*].

---

[*]https://github.com/mahmozaffari/VQACL_for_ICML26

