# OpenReview forum: "Calibrated Knowledge Aggregation in Bayesian Mixture-of-Experts for Continual VQA"
_ICML.cc/2026/Conference — ICML 2026 regular_

### Official Review · Reviewer_eEgz · 2026-03-10

**Soundness:** 3
**Presentation:** 2
**Significance:** 3
**Originality:** 2
**Overall Recommendation:** 4
**Confidence:** 4

**Summary:**

This paper studies continual VQA and argues that hard task-ID routing is suboptimal because non-native experts often answer queries better than the assigned task expert. It proposes a calibrated Bayesian mixture-of-experts with parameter-efficient task adapters, utility-driven soft routing, and Bayesian aggregation in a unified answer space. Experiments on VQA-v2 and TDIUC continual learning benchmarks show improved accuracy, reduced forgetting, and better calibration, demonstrating more reliable and transferable expert sharing.

**Compliance With Llm Reviewing Policy:**

Affirmed.

**Final Justification:**

The authors addressed my concerns about robustness. I tend to accept the paper.

**Key Questions For Authors:**

1. Could the model still produce overconfident wrong predictions when all top-k experts are insufficiently competent for a query, especially under severe distribution shift or ambiguous inputs?
2. How do inference speed and memory usage scale with the number of tasks, and do these costs become impractical when the expert pool grows large?

**Limitations:**

yes

**Strengths And Weaknesses:**

Strengths:
1. The method section is well organized and clearly explains the motivation, routing design, Bayesian aggregation, and calibration strategy in a coherent and easy-to-follow manner.
2. The experimental section is thorough, with strong analyses of transfer, forgetting, calibration, and routing behavior, which makes the empirical findings convincing and informative.



Weakness:
1. Evaluating only relatively small VQA models is not fully convincing; it would be more meaningful to test the method on stronger MLLM backbones such as LLaVA.
2. In the era of large multimodal models, the practical importance of studying continual learning specifically for traditional VQA settings should be better justified.
3. The authors should vary the task sequence in the continual learning benchmarks to examine how sensitive the proposed method is to ordering effects.

---

> ### Author Rebuttal · Authors · 2026-03-31
>
> ## Response to W1: Small VQA Models Evaluated
> We add experiments with BLIP (VQA-v2 dataset) and Qwen3-VL-4B (continual medical VQA). (Full discussions at :[Link](https://bit.ly/4rY5knr))
>
> **Table 1: BLIP on VQA-v2**
>
> | Method | Acc | Forgetting | ECE |
> |---|---|---|---|
> | zero-shot | 75.42 | | | 0.33 |
> | naive sequential LoRA | 78.03 | 4.57 | 0.29 |
> | + Hard routing | 79.19 | 1.60 | 0.29 |
> | + Bayesian (ours) | 80.49 | 1.26 | 0.17 |
>
> **Table 2: Qwen3-VL-4B on continual medical VQA**
>
> | Method | Acc | Forgetting | ECE |
> |---|---|---|---|
> | zero-shot | 67.09 | | | 0.32 |
> | Naive sequential LoRA | 71.50 | 6.89 | 0.28 |
> | + Hard routing | 77.22 | 0.53 | 0.13 |
> | + Bayesian (ours) | 77.47 | 0.39 | 0.09 |
>
> Two takeaways: (1) Stronger pretraining does not eliminate continual forgetting: Naive LoRA still causes
> forgetting (4.57 for BLIP, and 6.89 for Qwen3-vl-4B); our method reduces forgetting significantly on both models. This is more consequential for large VLMs, where forgetting expensive pretrained knowledge is costly to recover. (2) The Bayesian variant also improves calibration (ECE: 0.13→0.09 on MedVQA, and 0.29→0.17 on VQA-v2), critical in sensitive domains like medical VQA. Our method is thus complementary to, not competing with, large VLMs.
>
> ---
>
> ## Response to W2: Practical Importance of CL for Traditional VQA
>
> We agree this motivation should be stated more clearly. Our goal is not to restrict CL to traditional VQA; we use classification-style VQA as a controlled testbed for a broader problem also present in MLLMs: continual multimodal adaptation under overlapping tasks, shifting domains, and unreliable hard routing. This setting remains practically relevant: (1) many real deployments (on-device, latency-constrained, privacy-sensitive) cannot repeatedly retrain large models; (2) structured VQA is preferable in regulated
> applications for its controllable, auditable outputs; (3) a unified answer space enables direct calibration evaluation, isolating our core contribution. A new experiment with Qwen3-VL-4B on 4-task MedVQA confirms transfer: naive training yields substantial forgetting (acc: 71.50, forgetting: 6.89), our method reduces forgetting, and Bayesian routing improves calibration over hard routing.
>
> ---
>
> ## Response to W3: Task Ordering Sensitivity
> We evaluated across 5 orderings per dataset (mean ± std):
>
> | Dataset | Method | Acc (%) | Forget. (%) | ECE |
> |---|---|---|---|---|
> | VQA-v2 | Bayesian | 56.1 ± 0.16 | 0.3 ± 0.12 | 0.095 ± 0.006 |
> | VQA-v2 | Hard route | 55.6 ± 0.09 | 0.6 ± 0.16 | 0.145 ± 0.014 |
> | TDIUC-Q | Bayesian | 68.4 ± 0.07 | 0.5 ± 0.13 | 0.159 ± 0.001 |
> | TDIUC-Q | Hard route | 68.3 ± 0.15 | 0.6 ± 0.25 | 0.166 ± 0.002 |
> | TDIUC-V | Bayesian | 84.0 ± 0.19 | 3.3 ± 0.23 | 0.055 ± 0.004 |
> | TDIUC-V | Hard route | 82.5 ± 0.44 | 4.5 ± 0.52 | 0.134 ± 0.003 |
>
> Our method consistently outperforms hard routing across all orderings with lower variance. For per-task accuracy breakdowns across all orderings for each dataset, see [full results](https://bit.ly/4tjHns2).
>
> ---
>
> ## Response to Q1: Overconfident Predictions Under Insufficient Competence
>
> To test overconfident failures, we evaluate on 5 held-out TDIUC-Q categories as OOD tasks. The results
> reveal three regimes: when partial expert competence exists (utility affordance), Bayesian aggregation improves both accuracy and ECE by pooling transferable evidence across experts; when experts agree on the
> correct answer region but differ in confidence (object presence, sentiment understanding), aggregation preserves accuracy while reducing overconfidence; and when all experts lack competence (absurd ), ECE still
> marginally improves by preventing a single overconfident expert from dominating. The one exception is
> sport recognition, where aggregation dilutes a weak but unique correct signal from a single expert, slightly
> worsening both metrics, confirming that Bayesian aggregation is most effective under meaningful routing
> uncertainty, and can mildly hurt when one expert is uniquely correct but underweighted by the router.
>
> | Category | ECE Hard | ECE (Bayes) | Acc (Std) | Acc (Bayes) |
> |---|---|---|---|---|
> | utility\_afford. | 0.31 | **0.24** | 13.45 | **16.96** |
> | object\_presence | 0.40 | **0.37** | **49.9** | 49.8 |
> | sentiment\_understand. | 0.44 | **0.42** | **40.85** | 40.69 |
> | absurd | 0.73 | **0.70** | 0.0 | 0.0 |
> | sport\_recognition | **0.471** | 0.474 | **0.09** | 0.0 |
>
> ---
>
> ## Response to Q2: Inference Speed and Memory Scaling
>
> Inference cost is O(k): a lightweight 2-layer router selects top-k experts (k=3), so only those experts execute.
> Bayesian adds a small constant overhead (top-3 vs. top-1) over hard routing, not full pool evaluation.
> Memory grows linearly but stays lightweight: the shared ViLT backbone (111.6M params, 425.7 MiB)
> is frozen once; all 10 experts together add only 39.5M params (150.7 MiB, ≈35% overhead), giving a 576.4
> MiB final model. Expert merging or pruning are natural extensions for very large pools.

---

> > ### Author Rebuttal · Reviewer_eEgz · 2026-04-03
> >
> > The rebuttal meaningfully improves the paper by adding experiments on stronger backbones and multiple task orderings, which address part of my empirical concerns. However, I remain only partially convinced on the broader practical significance beyond controlled VQA settings, and on scalability/robustness when expert competence is uniformly weak or the expert pool grows substantially. Thus, while the paper has clear merits, I do not think the rebuttal fully resolves my main concerns.

---

> > > ### Author Response · Authors · 2026-04-07
> > >
> > > We thank the reviewer for the constructive feedback and for recognizing the improvements made during rebuttal. We address the remaining concerns below.
> > >
> > > ## Robustness
> > >
> > > To probe robustness when expert competence is degraded under distribution shift, we evaluate on AdVQA, an adversarial VQA dataset. This provides a fuller picture than our earlier held-out TDIUC-Q analysis, which did not span all overlap regimes.
> > > Across models trained on all datasets, overall accuracy on AdVQA is expectedly low, but Bayesian aggregation consistently preserves or slightly improves accuracy relative to hard routing
> > > (mean 26.13 → 26.57) while substantially improving calibration (mean ECE 0.41 → 0.34). Calibration gains are largest for VQA-v2 and TDIUC-V, where expert overlap is stronger, and smaller but still positive for TDIUC-Q, where experts are more disjoint. This matches our central claim: Bayesian aggregation is most effective when multiple experts remain plausibly relevant, while under broader weak-competence adversarial regimes its main benefit is reducing overconfidence rather than producing large accuracy gains.
> > >
> > > | Train Set | Hard Acc / ECE | Bayes Acc / ECE |
> > > |---|---|---|
> > > | VQA-v2 | 32.83 / 0.26 | 32.85 / 0.20 |
> > > | TDIUC-V | 24.13 / 0.48 | 24.90 / 0.36 |
> > > | TDIUC-Q | 21.42 / 0.50 | 21.97 / 0.48 |
> > >
> > > ### qualitative examples:
> > >
> > > **Q:** "What is this animal called when it is under 1 year old?" — *GT: kitten*
> > >
> > > | Expert | w | Conf | Pred | | Method | Answer | Conf  |
> > > |---|---|---|---|---|---|---|---|
> > > | E0 | 0.26 | 0.98 | cat | | Hard | cat | 0.98 |
> > > | E8 | 0.22 | 0.06 | yes | | Bayesian | cat | 0.42 |
> > > | E9 | 0.15 | 0.00 | yes |
> > > | E2 | 0.12 | 0.51 | no |
> > > | E5 | 0.08 | 0.05 | nothing |
> > >
> > > Both methods predict ”cat” incorrectly (GT: ”kitten”). Hard routing commits to E0 with a near-maximum overconfidence despite being wrong. Bayesian, lacking any competent expert, spreads weight broadly across experts (router
> > > entropy=1.96), reducing confidence to 0.42. Full discussion and examples at:
> > > [Link](https://bit.ly/ood-ex)
> > >
> > > ---
> > >
> > > ## Scalability
> > >
> > > We thank the reviewer for raising this important concern. While the main contribution of the paper is the
> > > routing-and-aggregation mechanism, the reviewer is right that the default one-expert-per-task grows linearly with the task stream. In longer streams, maintaining a separate expert for all tasks may become impractical even with lightweight experts.
> > >
> > > To test whether this growth can be moderated without changing the core method, we implemented a
> > > lightweight reuse-before-expand extension as a proof-of-concept. For each new task, we first use a held-out
> > > split to compute mean router mass assigned to prior slots, and keep the top-3 adapter slots as
> > > reuse candidates. After aligning the new task labels through the unified answer vocabulary, we compare
> > > three reuse modes against *full expand*: *full reuse* (remap the parent logits; no new parameters), *residual reuse* (reuse the frozen slot, learn only a small low-rank residual for label shift / novel labels), and *fresh head reuse* (reuse the adapter slot, attach new task head). Trainable candidates are probed for 2 epochs by updating just the lightweight head/residual parameters while keeping reused slot fixed; full reuse is evaluated with no additional training. We then choose the smallest-param. candidate with held-out accuracy within $\delta$= 0.005 of best candidate. So, a new slot is created only when reuse causes a meaningful loss.
> > >
> > > On ViLT/VQA-v2, reuse is selected in 4 of the 9 eligible stages (first task excluded): 2
> > > full and 2 residual reuse (fresh head reuse is never selected). Concretely, location, type,
> > > and subcategory reuse the recognition slot, and causal reuses commonsense slot, yielding 6 active
> > > slots instead of 10. Table below shows, this substantially reduces added parameters while keeping
> > > performance essentially stable. Relative to full expansion, accuracy and forgetting remain very
> > > similar. Under reuse, Hard routing and Bayesian are tied in accuracy, but Bayesian still retains the clear calibration advantage (ECE 0.10 vs. 0.15), so the reliability benefit of calibrated aggregation persists. Its final ECE is somewhat higher than the
> > > original Bayesian model (0.07), which is plausible since sharing slots across tasks can make
> > > the reused experts slightly less specialized/calibrated than fully expanded experts. Overall, results
> > > suggest that a substantial fraction of tasks can be absorbed into existing slots with little practical
> > > loss, significantly reducing long-run memory growth.
> > >
> > > | Variant | Added Params | Hard Acc  / ECE | Bayes Acc / ECE |
> > > |---|---|---|---|
> > > | Full expand | 41.7M | 63.92 / 0.13 | 64.16 / 0.07 |
> > > | Reuse-before-expand | 23.3M | 65.01 / 0.15 | 64.97 / 0.10 |
> > >
> > > Main trade-off is the short probing stage at task arrival: compared with full-expansion, reuse selection adds a small one-time training cost. However, once reuse is selected, the model permanently maintains fewer slots, reducing long-run parameter and memory growth.

---

### Official Review · Reviewer_Dhbu · 2026-03-11

**Soundness:** 3
**Presentation:** 3
**Significance:** 3
**Originality:** 3
**Overall Recommendation:** 5
**Confidence:** 3

**Summary:**

This paper proposes a calibrated Bayesian Mixture-of-Experts framework for continual VQA to address the limitations of traditional "hard routing." Instead of relying on task-ID supervision, the authors use utility-driven routing and entropy regularization to identify the most capable experts across overlapping tasks. By performing Bayesian aggregation in a unified answer space, the model effectively pools evidence from multiple experts, reducing overconfidence and catastrophic forgetting. Experimental results demonstrate the proposed methods.

**Compliance With Llm Reviewing Policy:**

Affirmed.

**Final Justification:**

The authors' reply addresses my concerns.

**Key Questions For Authors:**

- Potential limited scalability of the Unified Answer Space. The proposed Unified Answer Space Projection relies on a pre-defined, global dictionary that must be maintained and updated as new tasks arrive. This may limit the real-world application of proposed framework and arise several concerns such as open-set vulnerability, memory overhead. Could authors provide clarification on this issue?
- Insufficient sensitivity analysis of the entropy regularization hyperparameter. While the authors demonstrate that entropy regularization effectively prevents the router from collapsing into a "hard-routing" mode, the submission lacks a detailed sensitivity analysis regarding the regularization hyperparameter λ.
- Typos. The formula in section 4.3 has two overlapping indices.

**Limitations:**

The limitation is not discussed.

**Strengths And Weaknesses:**

- The motivation is quite clear.
- This paper is well-written and easy to follow.
- The dynamic routing with MoE for continual learning is interesting.
- The experiments are extensive and demonstrate the effectiveness.

---

> ### Author Rebuttal · Authors · 2026-03-31
>
> ## Response to Question 1: Scalability of Unified Answer Space
>
> Thank you for raising this point. We agree that the role and scalability of the unified answer space should be
> clarified more explicitly. Our method does not assume a fixed global dictionary known in advance. Rather,
> at continual stage t, the shared answer space is simply the union of answers observed so far, and each new
> task contributes only its task-local answer set together with a lightweight local-to-global alignment map.
> Previously learned experts remain unchanged. The purpose of this shared space is not to introduce an
> additional restriction, but to make Bayesian aggregation well-defined when different experts have different
> task-local output heads.
>
> Importantly, the size of this shared vocabulary grows with the number of genuinely new answers, not
> with the number of tasks. Our appendix shows that answer overlap is substantial in the settings where our
> method is most effective (notably VQA-v2 and TDIUC CL-VS), while gains are smaller in the more separable
> TDIUC CL-LS regime. Thus, in the overlap-heavy scenarios most relevant to our Bayesian aggregation, the
> cumulative shared vocabulary grows much more slowly than the sum of task-local vocabularies.
> We also note that Eq. (12) is written as a matrix for clarity, but it need not be a dense matrix in practice.
> Since each local answer maps to exactly one shared answer index, the projection can be stored as a sparse
> index map, so its memory overhead is negligible relative to the task-specific adapters and heads.
> Finally, we agree that handling truly novel answers beyond the support of the currently available task
> heads is an important challenge. However, this vulnerability is not introduced by the unified projection itself;
> it is inherited from the benchmark formulation used by classifier-style continual VQA methods. Importantly,
> the core idea of our paper is not restricted to this instantiation. In the rebuttal, we additionally provide
> results on generative MLLM backbones (Qwen-3vl-4B and BLIP), where answers are generated rather than
> selected from a predefined answer list, and observe the same qualitative benefits from our routing and
> aggregation strategy.
>
> ---
>
> ## Response to Question 2: Sensitivity Analysis of Entropy Regularization λ
> We provide a detailed sensitivity analysis of λ in Appendix J.2 (Figure 10), sweeping λ ∈ {0, 0.1, 0.2, 0.5, 1.0},
> on VQA-v2 (ViLT). Since the utility objective is linear over the simplex, λ → 0 drives near one-hot routing;
> increasing λ counteracts this collapse and preserves meaningful uncertainty over multiple experts, which is
> critical for Bayesian aggregation. We observe a clear accuracy–ECE trade-off: moderate values (λ ≈ 0.1–
> 0.2) improve *both* accuracy and calibration (Bayesian accuracy: 63.56 → 64.16; ECE: 0.141 → 0.074),
> while larger values (λ ≥ 0.5) cause over-smoothing: the router is forced to maintain high entropy even when
> one expert is clearly superior, diluting strong predictions and injecting noise into the aggregated posterior
> (accuracy drops to 62.68, ECE degrades from 0.074 to 0.107, losing most of the calibration benefit). We use λ = 0.1 throughout our experiments.
>
> Additionally, we provide sensitivity analyses for two other key hyperparameters: memory buffer size for
> router training (Appendix J.1), and top-k for Bayesian aggregation (Appendix J.3).
>
> ---
>
> ## Response to Q3: Typos / Overlapping Indices
>
> Thank you for catching this. We have corrected the overlapping indices in the formula in Section 4.3 in the revised manuscript.
>
> ---
>
> ## Response to Limitations
>
> We thank the reviewer for raising this. Limitations are discussed in Appendix L; we agree they deserve more visibility and will add a pointer in the main paper. Briefly, we identify three key limitations: (1) calibration gains from Bayesian aggregation are strongly mediated by task overlap i.e. in near-separable streams the router collapses to single-expert behavior, leaving limited headroom for improvement; (2) aggregation is bounded by per-expert reliability, as experts trained in isolation without calibration-aware objectives can remain miscalibrated on non-native inputs, which Bayesian averaging may inherit; (3) our framework assumes access to task boundaries during training, and relaxing this toward task-free expert creation remains important future work.

---

> > ### Author Rebuttal · Reviewer_Dhbu · 2026-04-03
> >
> > Thanks the authors for the clarification. My concerns are addressed and I will increase my score.

---

> > > ### Author Response · Authors · 2026-04-03
> > >
> > > We thank the reviewer for confirming that we have addressed all their concerns, and for their time and valuable insights. We appreciate the positive assessment and support for acceptance. We have incorporated the additional discussion about scalability of the unified answer spaces in the revised paper.

---

### Official Review · Reviewer_Hmdb · 2026-03-13

**Soundness:** 3
**Presentation:** 2
**Significance:** 2
**Originality:** 3
**Overall Recommendation:** 4
**Confidence:** 2

**Summary:**

This paper proposes a multi-expert architecture to tackle continual VQA. The authors train an expert for each task and a router to aggregate answer predictions from multiple experts given a new question during inference. Experts share the same base VLM, adapted through several LoRA parameter sets and equipped with answer heads. The router is trained to maximize total expert performance while smoothing routing prediction. The experiments show improvement over previous continual learning methods and experts with oracle routing.

**Compliance With Llm Reviewing Policy:**

Affirmed.

**Final Justification:**

The experiments in the rebuttal show that the proposed method may benefit specialized and professional VQA with small models. This may bring positive impacts to the community, and thus I'm increasing my rating.

**Key Questions For Authors:**

Questions
1. Could the authors provide several example questions and the corresponding predictions from different experts?
2. Could the authors show comparisons with large VLMs?
3. Could the authors put the reference section in the main paper? This will greatly improve the reading experience.
4. I’m also curious whether more experts can help with a new question or introduce more noise. Could the authors show performance at different stages (numbers of trained experts)?

**Limitations:**

yes

**Strengths And Weaknesses:**

Strengths

The proposed method is intuitive and reasonable. Representing task-specific capabilities with LoRA and aggregating all learned capabilities for novel problems looks smart and efficient.

Weaknesses

1. The paper presentation does not look comfortable for me, especially the title spacing and the font size. In addition, there is no qualitative comparison or examples. This could tell us how different experts trained on different tasks contribute to a new question.
2. There is no comparison of state-of-the-art VQA systems, especially large VLMs. While I can understand it might be valuable to improve small models, such as generalizability or robustness to forgetting, I feel the authors may at least discuss the differences, benefits, or anything that makes the proposed system important in this era, and also show the performance gap.

---

> ### Author Rebuttal · Authors · 2026-03-31
>
> ## Response to W1: Presentation
>
> We will rebalance the title/header spacing on the first page, enlarge fonts in figures and tables, reduce crowding by redistributing dense result panels, shorten or simplify captions where possible, and add qualitative examples in the main paper to improve visual flow and readability.
>
> ## Response to W1/Q1: Qualitative Examples
> Three illustrative examples below (full examples: [Link](https://bit.ly/qual-ex)).
> - - -
> **Ex. 1 | task: 2 | Q: "Is the lady moving down slope?" | GT: yes***
>
> | Expert | w    | Conf | Pred  | ║ | Method          | Answer | Conf | Acc |
> |---|---|---|---|---|---|---|---|---|
> | E5     | 0.38 | 0.74 | "no"  | ║ | Hard    | "no"     | 0.73 | 0.0 |
> | E2     | 0.37 | 0.99 | "yes" | ║ | Bayesian (ours) | "yes"    | 0.62 | 1.0 |
> | E3     | 0.15 | 0.64 | "yes" | ║ |                 |        |      |     |
>
> Hard routing commits to E5 (w=0.38), predicting ”no” incorrectly. Bayesian aggregation pools evidence
> from E2 and E3, both predicting ”yes” with high confidence, overriding E5’s incorrect answer.
>
> ---
>
> **Ex. 2 | task: 2 | Q: "Is the bed made?" | GT: yes**
>
> | Expert | w | Conf | Pred |║| Method | Answer | Conf | Acc |
> |---|---|---|---|-|---|---|---|---|
> | E2 | 0.44 | 1.00 | "no" |║| Hard | "no" | 1.00 | 0.0 |
> | E3 | 0.20 | 0.81 | "yes" |║| Bayes. | "no" | 0.79 | 0.0 |
>
> Both fail, but hard routing is maximally overconfident (conf=1.00). Bayesian aggregation injects uncertainty from dissenting experts, reducing confidence to 0.79 (better calibrated even in failure).
>
> ---
>
> **Ex. 3 | task: 1 | Q: "Where is the dog laying down?" | GT: bed***
>
> | Expert | w    | Conf | Pred | ║ | Method     | Answer | Conf | Acc |
> |---|---|---|---|---|---|---|---|----|
> | E1     | 0.17 | 0.07 | "bed"  | ║ | Hard | "bed"    | 0.07 | 1.0 |
> | E0     | 0.14 | 0.87 | "bed"  | ║ | Bayes.     | "bed "   | 0.31 | 1.0 |
>
> Both correct. Hard routing relies solely on E1’s uncertain prediction (conf=0.07). Bayesian aggregation
> identifies corroborating evidence from E0 (conf=0.87), amplifying final confidence from 0.07 to 0.31.
>
> These examples demonstrate three behaviors: (1) non-native experts override misrouted decisions; (2)
> Bayesian pooling yields better-calibrated uncertainty even in failure; (3) corroborating experts amplify
> confidence on correct predictions.
>
> ## Response to W2/Q2: Large VLM Comparison ([Full discussion](https://bit.ly/4rY5knr))
> We add large-model experiments with BLIP on VQA-v2 and Qwen3-VL-4B on continual medical VQA.
> Two points emerge. Stronger pretraining does not remove the CL problem. On BLIP, naive LoRA fine-tuning still exhibits substantial forgetting (4.57). Our method preserves prior knowledge
> while improving accuracy:
>
> | Method | Acc | Forget | ECE |
> |---|---|---|---|
> | 0-shot | 75.42 | - | 0.33 |
> | naive (LoRA) | 78.03 | 4.57 | 0.29 |
> | + Hard | 79.19 | 1.60 | 0.29 |
> | + Bayesian | 80.49 | 1.26 | 0.17 |
>
> On Qwen3-VL-4B medical VQA, Bayesian aggregation improves both accuracy and calibration over hard
> routing:
>
> | Method | Acc | Forget | ECE |
> |---|---|---|---|
> | 0-shot | 67.09 | | | 0.32 |
> | Naive (LoRA) | 71.50 | 6.89 | 0.28 |
> | + Hard | 77.22 | 0.53 | 0.13 |
> | + Bayesian | 77.47 | 0.39 | 0.09 |
>
> This matters more, not less, for large VLMs. When a large model forgets, expensive pretrained knowledge is
> lost and retraining is impractical. Our method is complementary to stronger VLMs, not limited to smaller
> ones. Smaller backbones in the main paper provide a controlled setting to isolate routing, forgetting, and
> calibration effects; these results confirm the principles transfer to stronger models. Finally, small models
> remain practically relevant under efficiency, latency, deployment cost, or privacy constraints, and are often
> harder to calibrate precisely due to lower capacity, making our calibrated aggregation especially valuable in
> that regime. We will revise to discuss the performance gap and complementary role of our method explicitly.
>
> ---
>
> ## Response to Q4: Performance at Different Stages
>
> We report per-task accuracy and ECE at each stage for two representative tasks:
>
> | Task | | 1 | 2 | 3 | 4 | 5 | 6 | 7 | 8 | 9 | 10 |
> |---|---|---|---|---|---|---|---|---|---|---|---|
> | Recognition | Acc | 50.5 | 50.5 | 49.8 | 49.4 | 49.4 | 48.91 | 48.20 | 49.3 | 49.8 | 49.1|
> | | ECE | 0.09 | 0.1 | 0.1 | 0.1 | 0.09 | 0.1 | 0.09 | 0.08 | 0.05 | 0.05 |
> | Location | Acc | - | 30.2 | 23.9 | 25.0 | 23.4 | 23.3 | 23.0 | 24.4 | 30.5 | 27.9 |
> | | ECE | - | 0.07 | 0.1 | 0.09 | 0.09 | 0.1 | 0.09 | 0.06 | 0.03 | 0.05 |
>
> More experts do not cause progressive noise accumulation. Recognition stays flat (50.5→49.1) with ECE improving (0.09→0.05). Location varies across stages but shows bounded fluctuation rather than monotonic degradation, with ECE still improving overall (0.07→0.05). This is consistent with sparse top-k routing, which suppresses irrelevant experts while allowing the mixture over plausible experts to shift as the pool grows. Full curves: [link](https://bit.ly/stage-perf).

---

> > ### Author Rebuttal · Reviewer_Hmdb · 2026-04-03
> >
> > Most of my concerns have been resolved. While the method shows marginal improvement for BLIP on VQA v2 (which I believe is still outdated), it shows noticeable improvement on specialized problems. The method may benefit customized models for specialized or professional tasks that proprietary systems can struggle with.
> > I'm increasing my rating.

---

> > > ### Author Response · Authors · 2026-04-07
> > >
> > > We thank the reviewer for confirming that we have addressed most of their concerns, and for their time and valuable insights. We are glad that the potential of our method in specialized domains is recognized.
> > >
> > > The reviewer's feedback and questions have strengthened our revision. We appreciate the positive assessment and support for acceptance.

---

### Decision · Program_Chairs · 2026-04-30

**Decision:**

Accept (regular)

**Comment:**

Multi-expert architecture for reliability in continual VQA. Improves “hard (top-1 expert) routing by combining [pieces of] evidence of expert predictive distributions to recover from a) overconfident wrong predictions, b) underconfident correct predictions, 3) incorrect predictions due to misrouting.”

Rating: Reviewer Hmdb (4) , Reviewer Dhbu (5) , Reviewer eEgz (4). The reviewers find the paper well-written and well-motivated. They also find the proposed approach intuitive with sound+efficient aggregation schemes (Bayesian aggregation, LoRA utilization) with strong empirical results to back up the claims.

Major Concerns

1. [Reviewer Hmdb, Reviewer eEgz, resolved] Small base VQA model. The reviewers want to see larger VLMs to justify the proposed approach’s relevance. The authors provided BLIP (VQA-v2 dataset) and Qwen3-VL-4B (continual medical VQA) results during the rebuttal.

2. [Reviewer eEgz, somewhat resolved] Robustness to component changes. Reviewer eEgz questioned the sensitivity to task ordering and weak experts. They also questioned the costs of inference speed and memory usage as the expert pool grows large. The authors evaluated the framework across 5 task orderings. The authors also provided an additional result on adversarial AdVQA to test robustness. Finally, introduced a "reuse-before-expand" extension to address memory scaling, though Reviewer eEgz remained partially unconvinced regarding this part.

3. [Reviewer Dhbu, resolved] Scalability. Reviewer Dhbu questions the approach’s scalability due to a pre-defined global answer dictionary. The authors clarified that the unified answer space is the union of answers observed so far, which grows slowly under overlap-heavy scenarios.